# Emotional associative memory is disrupted by directed forgetting

Anastasia Chalkia [1✉], Niels Vanhasbroeck[2], Lukas Van Oudenhove [3,4], Merel Kindt [5] & Tom Beckers [1]

Memory is susceptible to voluntary disruption, for instance, through directed forgetting manipulations, in which people are cued to intentionally "forget" information. Until now, directed forgetting has been primarily studied for declarative memory performance. Here, we demonstrate that directed forgetting can also disrupt associative memories acquired through fear conditioning. In two experiments, participants showed poorer recognition and recall of images paired with electric shocks when instructed to forget, compared to when instructed to remember them. Further, they also showed weaker skin conductance responses to images paired with shocks that they were instructed to forget, despite repeated, full reinforcement of the aversive outcome. Our findings provide evidence for the effect of directed forgetting not only on declarative but also physiological read-outs of emotional memory, thereby suggesting that forgetting instructions can be applied to interfere with emotional associative memory.

[1] Centre for the Psychology of Learning and Experimental Psychopathology, Faculty of Psychology & Educational Sciences, KU Leuven, Leuven, Belgium. [2] Research Group of Quantitative Psychology and Individual Differences, Faculty of Psychology & Educational Sciences, KU Leuven, Leuven, Belgium. [3] Laboratory for Brain-Gut Axis Studies (LaBGAS), Translational Research Centre for Gastrointestinal Disorders (TARGID), Department of Chronic Diseases and Metabolism, KU Leuven, Leuven, Belgium. [4] Cognitive and Affective Neuroscience Lab, Department of Psychological and Brain Sciences, Dartmouth College, Hanover, NH, USA. [5] Department of Clinical Psychology, University of Amsterdam, Amsterdam, The Netherlands. ✉email: anastasia.chalkia@kuleuven.be

Optimal functioning requires a fine balance between remembering and forgetting. Too much forgetting is detrimental and a hallmark feature of pathologies such as Alzheimer's disease and other amnestic disorders, however, ample evidence suggests that some degree of forgetting is important for proper cognitive functioning (see e.g., refs. [1,2]). For instance, forgetting promotes enhanced emotion regulation, facilitates the acquisition of new knowledge, and ensures the maintenance of memory relevance[2]. Moreover, some clinical conditions are characterized by memory being excessively vivid and retrievable, as in the case of recurring flashbacks in post-traumatic stress disorder (PTSD) patients. Thus, while in memory research a great emphasis is placed on understanding and improving remembering, equally important is the quest for mechanisms and principles that allow us to forget trivial but also (negative) emotionally salient information.

One area of research that has focused on enhancing the transience of memory and unraveling its underlying mechanisms is the one concerned with directed forgetting (DF)[3], where researchers investigate the effects of forgetting instructions on declarative memory performance[4]. In the basic paradigm, isolated words (item-method) or lists of words (list-method) are presented, and they are followed by mnemonic cues: the remember (R) cue acts as an instruction to remember certain items, whereas the forget (F) cue indicates items to be forgotten[4]. While there are various strands in DF research, and multiple variations to the original paradigm, findings over the past 50 years have been very robust, with participants exhibiting impaired recall and recognition of items that they were instructed to forget during subsequent declarative memory testing—referred to as the DF effect[4]. Even though research on DF has been flourishing, no consensus has yet emerged regarding the underlying mechanism that may be responsible for this effect. Some of the proposed mechanisms are linked to the specific method of DF induction used[5,6]. For instance, item-method DF has most notably been interpreted in terms of selective rehearsal (i.e., subjects rehearse R items only, while F items decay)[7,8], retrieval inhibition (i.e., retrieval of F items is suppressed)[9,10] or attentional inhibition (i.e., engagement of attentional mechanisms during encoding to suppress the processing of F items)[11,12].

Research on DF typically focuses on encoding and disruption of single items, either presented in isolation or in lists, rather than on associative information. Yet remembering associations between stimuli is fundamental for cognitive functioning, e.g., because it allows for predictive learning[13]. In addition, associations are a key driver of memory retrieval, like when the sight of a dark alley reminds you of a violent assault. In recent years, researchers have started investigating DF of declarative and procedural read-outs of associative memories, employing either unrelated word pairs, scene-object pairs, or arbitrary stimulus-response (S-R) pairings (left/right key presses in response to words), and have observed a DF effect for such associative information as well[14–17], but see ref. [18]. However, in these studies the authors solely measured general recognition performance, and while they observed diminished recognition, in most cases it was impossible to disentangle whether memory deficits were due to an impaired recognition of the items (i.e., disrupted item recognition) or an impaired retention of the association between item pairs (i.e., disrupted associative recognition). Of note, DF has also been shown to affect explicit ratings in evaluative conditioning (EC; the change in valence of a stimulus as a result of it being paired with a clearly positive or negative stimulus), suggesting reduced associative memory for pairings followed by a forget instruction[19].

Yet these DF effects on declarative memory performance, as probed through verbal report, may have little bearing on the expression of emotional associative memory, considering that the expression of emotional memory is often more resistant to interference[20] and that it can be expressed through automatic reactions (e.g., psychophysiological responses). Indeed, DF procedures utilizing (single-item) emotional words or images have yielded weak DF effects in verbal report at best, with item recall and/or recognition for emotional stimuli consistently higher than for neutral stimuli[21–23]. No research has addressed the ability of DF to reduce non-verbal memory expression (e.g., psychophysiological responding), despite the central role of non-voluntary retrieval of associative memory information in emotional disorders.

In the lab, Pavlovian fear conditioning procedures can be used to install, manipulate, and modify emotional (fear) memories in humans. Typically, in a differential fear conditioning procedure, two neutral stimuli, such as pictures (conditioned stimuli (CSs)), are presented, and one of them (CS+) is repeatedly paired with an aversive outcome, such as an unpleasant electric stimulus (unconditioned stimulus (US)), while the other (CS−) is never paired with the US. After multiple pairings, the CS+ comes to elicit a conditioned [fear] response (CR)[24]. Here we introduce a trial-unique, differential fear conditioning procedure, in which 24 simple line drawings of objects were presented one at a time, and half of them were followed by a mild electric shock US (CS+), while the other half were not (CS−). Crucially, an acoustic F cue was presented after half of the CS+ and CS− trials, indicating that those trials were to be forgotten. Our procedure also allowed for the inclusion of a physiological measure of fear responding (skin conductance reactivity to the CSs; SCR) and all CS+ and CS− items were presented three times in order to increase the strength of the CS+/US and CS−/NoUS associations (see Fig. 1 for an illustration of the procedure and Methods for full methodological details). We assessed memory retention for all

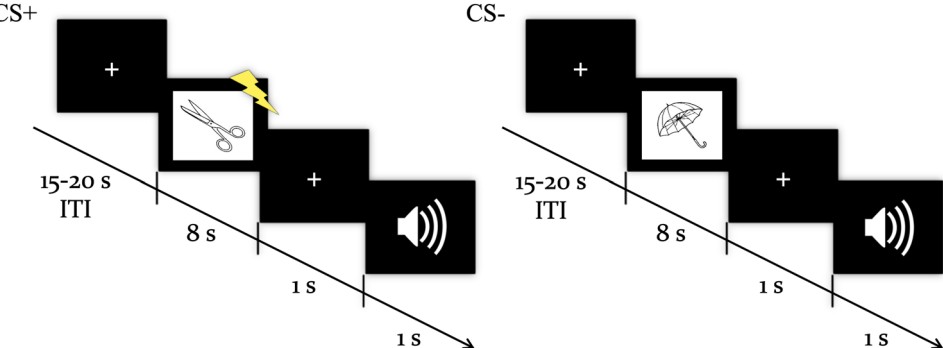

**Fig. 1 Illustration of the fear conditioning procedure.** All trials started with a fixation cross, followed by an image of a simple line object[25]. CS+ trials co-terminated with a 200-ms electric stimulus, while CS− trials were unreinforced. An acoustic tone that served as the cue to forget was presented 1 s following CS offset on half of the CS+ and half of the CS− trials.

items and their associations in subsequent free recall and recognition tasks and obtained retrospective liking ratings for all CS stimuli. We hypothesized that we would observe impaired recall and recognition following a forget instruction, as well as an attenuation of fear responding for items that were instructed to be forgotten (i.e., reduced SCRs to CS+ items that had been followed by the F cue). We also conducted a second experiment aimed at replicating our original results, and report both experiments here.

## Methods

**Preregistration**. All experimental procedures and planned statistical analyses were preregistered on AsPredicted (Experiment 1, preregistered on February 26, 2019: https://aspredicted.org/7x7b4.pdf; Experiment 2, preregistered on February 16, 2021: https://aspredicted.org/dd4xg.pdf). Additional analyses were performed beyond those that were preregistered. Specifically, for free recall and recognition, we had preregistered analyses only on the items that participants correctly categorized as CS+/CS− (i.e., on the corrected data), but we report additional analyses on the total items recalled/recognized (i.e., not accounting for errors in CS+/CS−categorization).

**Participants**. Students and community volunteers were recruited from the KU Leuven research pool for both experiments, which were conducted about 2 years apart. To establish the necessary sample needed for Experiment 1, we ran a small pilot study with 10 participants (not reported here) that yielded very large effect sizes (>1 for all outcome measures). When using these effect sizes in a power analysis, suggested sample sizes were very small ($N < 12$). To counter undue influence of the data of individual participants on the results, we preregistered a total sample size of 40. Out of 43 participants that we originally recruited, 1 was excluded for not following instructions (e.g., did not try to remember any of the items; see Procedure) and 2 because of technical malfunction (e.g., problematic SCR electrodes that did not correctly register responses), yielding the intended final sample of 40 participants (31 female participants), aged between 18 and 39 years ($M = 20.85$, $SD = 4.55$). To determine our sample size for Experiment 2, we conducted a power analysis using the effect size we obtained for our critical outcome of interest in Experiment 1 (i.e., the comparison between SCRs for CS+R and CS+F). Setting alpha at .05 and employing said effect size of $d = 0.50$, a sample of $N = 45$ in Experiment 2 should have yielded a power of 0.95 to detect this effect. We recruited 68 participants, 2 of whom necessitated exclusion due to technical malfunctions with the psychophysiological equipment. A further 21 participants were excluded for SCR non-responding (SCR amplitudes <0.02 μS on 75% of all trials), an exclusion criterion that we preregistered for the second experiment only. The final sample thus included 45 participants (28 female participants), aged between 18 and 27 years ($M = 20.82$, $SD = 2.70$). Given that the 23 excluded participants were fully tested but subsequently excluded solely for reasons relating to their physiology, we retained them in the final sample for the declarative memory tests (but we also report those outcomes with the preregistered $N = 45$ in the Supplementary Results). The final sample including all 68 participants (48 female participants), was aged between 18 and 42 years ($M = 21.44$, $SD = 4.04$). In both experiments, and in accordance with local ethics committee regulations, participants were first screened to be free from certain medical conditions, including pregnancy, cardiovascular/pulmonary/neurological/psychiatric or other serious medical conditions, presence of an electronic implant, pain at the hand or wrist, hearing problems, or a request from a physician to avoid stressful situations. They were further asked to self-report their sex at birth. We did not collect information on race or ethnicity. All participants gave written informed consent prior to the start of the studies and were compensated with partial research credits or a small monetary amount for their participation (Experiment 1: €8, Experiment 2: €16). Experimental procedures were approved by the KU Leuven Social and Societal Ethics Committee and were carried out in accordance with the Code of Ethics of the World Medical Association (Declaration of Helsinki).

**Conditioned stimuli (CSs)**. Forty simple line drawings were selected from the Snodgrass and Vanderwart[25] standardized set of images that contains exemplar objects from various categories (e.g., items of furniture, fruit, kitchen utensils, musical instruments, etc.). We selected objects from all categories except for the animal categories, as to avoid generating higher responding in relation to some of those pictures should our sample have included people with specific animal phobias. Additionally, we selected objects that scored high on name agreement (i.e., "spoon" which only has one name, rather than "road" which can also be referred to as "street," "lane," "alley," "avenue," "highway," etc.) in order to facilitate free recall scoring. For a complete list of all the selected items, please refer to the Supplementary Methods. In both experiments, the exact same images were used: 4 for the practice trials, 24 for the acquisition phase, and 12 novel ones for the recognition task. Across experiments, the same images were assigned to "practice," "acquisition," and "recognition" stimuli, but allocation of acquisition stimuli into stimulus type categories (CS+R, CS+F, CS−R, and CS−F) was completely random for all participants, meaning that an image that served as a CS+F for one participant could serve as a CS−R for another. The order of trial presentations was random for each participant, with the restriction that no trial type was presented more than twice in a row.

**Unconditioned stimulus (US)**. The US was a mild, 200-ms electric stimulus that was generated using a DS7A constant-current stimulator (Digitimer, Hertfordshire, UK). The electric stimulus was delivered to the top of the wrist of the dominant hand using a stimulating bar electrode (Digitimer), composed of two 8-mm stainless steel electrodes with an inter-electrode distance of 30 mm. Participants selected their own US intensity using a work-up procedure and were asked to settle for a level that was "uncomfortable, but not painful" (see Supplementary Results for average selected US intensities).

**Forget (F) cue**. A 16-kHz computer tone with a 1-s duration served as the F cue. It was presented binaurally through headphones (Sennheiser HD 202, Wedemark, Germany) at 90 dBA.

**Liking ratings**. Retrospective liking ratings were obtained for each CS during both experiments. All the images from the acquisition phase were presented at the center of the screen, one by one, and participants were asked to indicate how much they liked or disliked each image using an 11-point rating scale ranging from −5 "extremely dislike" to 5 "extremely like."

**Questionnaires**. In Experiment 2 only, participants completed certain personality questionnaires to probe for individual differences that may be related to DF. To assess mind-wandering, the Mind-Wandering Questionnaire (MWQ)[26] was used, which consisted of 5 items rated on a 6-point Likert scale ranging from "almost never" to "almost always." The Frost Multidimensional Perfectionism Scale—Brief (F-MPS-B)[27] was administered to evaluate perfectionism and consisted of 8 items, rated on 5-point Likert scale, ranging from "strongly disagree" to "strongly agree."

Last, the 8-item neuroticism scale from the Big Five Inventory (BFI)[28] was employed to measure neuroticism. Its rating was similar as for the F-MPS-B.

**Working memory tasks.** *N*-back tasks are commonly used to measure different aspects of working memory and executive functioning[29]. In Experiment 2, we employed a 2-back task as a measure of working memory updating[30]. During the task, a series of individual letters was presented on the screen, and participants had to respond (by pressing J on the keyboard) every time the current letter was the same as the letter that was presented 2 positions back. They were instructed to respond as fast as possible while trying not to make any mistakes and were allowed 3 s to respond before the next trial began. Participants completed a practice block of 50 trials (30 hit-trials) and two experimental blocks of 100 trials (30 hit-trials), all separated by 20-s inter-block intervals. Feedback was offered on all practice trials (i.e., "correct"/"missed"), but during experimental trials, feedback was only provided on missed trials. Reaction times (RT) were recorded for all responses and participants' accuracy (ACC) was recorded as number of hits, misses, false alarms, and correct rejections. Further, we computed the hit rate [=hits/(hits + misses)] and false alarm rate [=false alarms/(false alarms + correct rejections)] which were then used to obtain the dependent variable used in our analysis, $d'$ [$d' = z(H) - z(F)$, where $z(H)$ and $z(F)$ are the $z$ transformations of the hit rate and false alarm, respectively]. In case of perfect hit rates (1) or zero false alarm rates (0), we corrected $d'$ by replacing the 1 or 0 by $(N-0.5)/N$ or $0.5/N$, respectively, where $N = 200$, the number of total trials. The task was presented through Psychopy software[31].

Also just in Experiment 2, a Flanker task[32] was used to measure working memory inhibition. Participants were presented with a string of seven numbers on the screen and were instructed to respond based on the target number in the middle of the string. If the target was 1 or 2, they had to press F on the keyboard, and if the target was 3 or 4, they had to press J on the keyboard. The other six numbers surrounding the target were distractors and could form congruent strings (e.g., 1111111, 1112111; target and distractor trigger the same response) or incongruent strings (e.g., 1114111; target and distractor trigger the opposite response). Once a response was recorded, the next trial began. Participants completed 15 practice trials (which included feedback), followed by a block of 120 experimental trials (60 congruent trials; did not include feedback). A fixation cross was presented in-between trials for 1 s. The dependent variable for our analysis was the RT difference between congruent and incongruent trials.

**Skin conductance (SCR).** SCR was recorded using an isolated skin conductance coupler (LabLinc v71-23, Coulbourn Instruments, Allentown, PA) and two pre-gelled, disposable 11-mm Ag/AgCl electrodes (EL507, Biopac Systems, Goleta, CA) attached to the palm of the non-dominant hand. The SCR signal was measured at 1000 Hz and digitized online using a 16-bit AD converter (National Instruments NI-6221, Austin, TX). Offline data extraction was completed with a custom-made MATLAB toolbox (R2021a, MathWorks, Natick, MA). SCR amplitudes were determined by subtracting the average of a 2-s baseline (prior to CS onset) from the maximum response in a 0 – 7.5 s window following CS onset. All responses were kept in the analysis, and SCR data were $z$-transformed using the mean and standard deviation of all responses obtained. After transformation, any observation with a $z$-score above 4 or below −4 was defined as an outlier and replaced by linear trend at point using IBM SPSS Statistics 27.

**Procedure.** Experiment 1 began with asking participants for written informed consent and screening them for medical exclusion criteria. After attachment of electrodes, a shock work-up procedure was used to allow participants to select their own US intensity. The experimental task then started, and participants were informed that they would see different images of objects and that some would be paired with the US (CS+), while others would not (CS−). They were instructed that their task was to memorize the objects they saw and whether or not they were paired with the US, as a memory test would follow. Further, they were also told that the computer would randomly select some trials that would not be tested later; those trials would be followed by a sound from the headphones. When this sound was heard, participants did not need to try to remember the object they saw as it would allegedly not be tested later. The experiment began with 4 practice trials (one of each category) that were followed by an explanation (i.e., "You should remember that the robot was paired with the US, while the skateboard was not. You do not need to try to remember the saw and the mask."). At that time, they were allowed to ask the researcher questions if they did not understand the task. The acquisition phase followed, consisting of 24 individual images (12 CS+, 12 CS−), presented 3 times in blocks of 24 stimuli (all stimuli were shown once in each block). The acquisition phase began with a 5-min SCR habituation period and each trial began with the presentation of a fixation cross (500 ms in duration), followed by the CS that was presented for 8 s. On all CS+ trials, the US was introduced 7800 ms after CS+ onset and co-terminated with CS+ offset. One second after CS offset, the F cue was presented on 50% of all CS trials and followed by a 15–20 s inter-trial interval (ITI), where the fixation cross was presented again. Stimulus presentation and data acquisition were controlled through Affect 4.0[33], a custom-made, freely available software package for behavioral experiments.

Following the acquisition phase, participants were instructed to freely recall as many CS+/CS− items as possible, even the ones that had been followed by the F cue. They were allowed 4 min to write down all the items they could recall on a piece of paper and were asked to circle the ones that had been paired with the US. After the free recall task, a recognition task was conducted, in which 12 of the CS items presented during acquisition and 12 novel items were displayed. On each trial of the recognition task, participants saw an item on the screen along with the question: "Do you recognize this object as something you saw earlier?" If they answered "yes," a second question was presented: "Was it paired with the US?" If they answered "no" to the first question, the next trial immediately followed. After the recognition task was completed, all acquisition items were presented once again, and participants were asked to give their retrospective liking ratings.

In Experiment 2, the procedure described above was adhered to, however, some additional measures were also introduced. Prior to the attachment of electrodes and shock work-up procedure, participants completed the MWQ, the F-MPS-B, and the BFI-neuroticism on the computer. Further, after the conclusion of the DF part of the protocol, participants proceeded to a 10-min break, during which they were instructed to sit silently in the lab without doing anything. After the break, they proceeded to the working memory tasks, completing the *n*-back first, followed by the Flanker task. All other aspects of the procedure were identical to that of Experiment 1.

**Statistical analyses.** For both experiments, free recall and recognition data were treated in the same manner. First, we analyzed the total number of items recalled/recognized, irrespective of possible stimulus (CS+/CS−) categorization errors, using $2 \times 2$ repeated-measures (rm) ANOVAs with Stimulus

(CS+, CS−) and Instruction (Remember, Forget) as within-subject factors. Next, we analyzed the proportion of associations correctly categorized for those items correctly recalled/recognized (i.e., items correctly classified as CS+ or CS− divided by total items correctly recalled or recognized) by subjecting the recall/recognition data to similar $2 \times 2$ rm-ANOVAs. Follow-up, two-tailed $t$-tests were conducted to compare the total items recalled/recognized and the associations that were correctly categorized: CS+R versus CS+F and CS−R versus CS−F. For the liking ratings, we executed identical $2 \times 2$ (Stimulus × Instruction) rm-ANOVAs and $t$-tests across experiments.

SCR responses during blocks 2 and 3 of acquisition were averaged per stimulus type (CS+R, CS+F, CS−R, CS−F) for the main analysis. SCR data from the first block were not included in the main analysis, given that in our trial-unique procedure, differential SCR responding could not have developed prior to the second presentation of a given item. SCR data were subjected to $2 \times 2$ rm-ANOVAs with Stimulus (CS+, CS−) and Instruction (Remember, Forget) as within-subject factors. As a secondary analysis, we examined the last two blocks of acquisition separately, including Block (2, 3) as an additional factor in the ANOVAs. Planned follow-up two-tailed $t$-tests were conducted to compare average SCR responding during to-be-remembered CS+ items and to-be-forgotten CS+ items and to compare CS−R and CS−F items. In addition to the preregistered $t$-tests, non-preregistered two-tailed $t$-tests were performed to compare SCR during to-be-remembered CS+ versus CS− items and to-be-forgotten CS+ versus CS− items.

Finally, in Experiment 2, we investigated the relationship between certain individual difference factors and the magnitude of the DF effect. First, we calculated differential DF indices (DF Index = Remember – Forget) for the corrected recognition and free recall data, as well as two indices for SCR, one matching the previously introduced DF index (SCR Index = Remember – Forget) and the other examining only differences in CS+ responding (SCR CS+ Index = CS+R – CS+F). We then computed Pearson correlations between these indices and our dependent variables from the $n$-back and Flanker tasks, as was as the scores on the personality questionnaires. Alpha was set at 0.05 for all analyses, which were performed using JASP version 0.17.1[34]. All follow-up $t$-tests were Bonferroni-corrected as to maintain an identical analysis plan between the two experiments, even though a Bonferroni correction was only preregistered for Experiment 2. The assumption of sphericity was explicitly tested in ANOVAs with more than 3 levels of a repeated measure. Given the within-subjects design, a normal distribution and equal variances were assumed for all ANOVAs, but these assumptions were not formally tested.

**Reporting summary.** Further information on research design is available in the Nature Portfolio Reporting Summary linked to this article.

## Results

**SCR.** In Experiment 1, as expected, participants exhibited stronger SCR responses to CS+ than CS− items (main effect of stimulus, $F(1, 39) = 27.44$, $p < 0.001$, $\eta_p^2 = 0.41$, 95% CI [0.17, 0.58]), indicating successful fear acquisition. We found no evidence for a statistically significant difference between SCRs of remember and forget associations (main effect of instruction, $F(1, 39) = 2.15$, $p = 0.15$, $\eta_p^2 = 0.05$, 95% CI [0.00, 0.22]). Importantly, we found a significant stimulus by instruction interaction ($F(1, 39) = 13.67$, $p < 0.001$, $\eta_p^2 = 0.26$, 95% CI [0.06, 0.45]) (see Fig. 2a). Planned follow-up $t$-tests indicated lower SCRs to CS+ items that had been followed by a forget cue than to CS+ items that had not been followed by a forget cue ($t(39) = 3.49$, $p = 0.005$, $d = 0.74$, 95% CI [0.14, 1.34]), whereas there was no evidence for a significant difference between remember and forget CS− trials ($t(39) = -1.22$, $p = 1$, $d = -0.26$, 95% CI [−0.83, 0.31]). Further, in addition to the preregistered comparisons above, we also performed non-preregistered Bonferroni-corrected $t$-tests comparing responding to CS+ and CS− items separately for the remember and forget conditions. Fear learning was evident for remember trials, as to-be-remembered CS+ trials elicited stronger SCRs than to-be-remembered CS− trials ($t(39) = 6.41$, $p < 0.001$, $d = 1.50$, 95% CI [0.87, 2.13]), but not for forget trials, as we found no statistically significant difference between SCRs during CS+ forget and CS− forget trials ($t(39) = 2.13$, $p = 0.22$, $d = 0.50$, 95% CI [−0.15, 1.14]). In a secondary analysis, when entering Block (2, 3) as an additional factor in the ANOVA, the pattern of results remained the same. We retained the significant stimulus by instruction interaction ($F(1, 39) = 15.00$, $p < 0.001$, $\eta_p^2 = 0.28$, 95% CI [0.07, 0.47]) and main effect of stimulus ($F(1, 39) = 27.55$, $p < 0.001$, $\eta_p^2 = 0.41$, 95% CI [0.18, 0.58]), but main effects of instruction ($F(1, 39) = 2.12$, $p = 0.15$, $\eta_p^2 = 0.05$, 95% CI [0.00, 0.22]) and block ($F(1, 39) = 0.64$, $p = 0.43$, $\eta_p^2 = 0.02$, 95% CI [0.00, 0.16]) were not statistically significant, and neither was the interaction between stimulus, instruction, and block ($F(1, 39) = 0.04$, $p = 0.84$, $\eta_p^2 = 0.001$, 95% CI [0.00, 0.04]) (see Supplementary Results for additional analyses examining the development of SCR across the experiment, including all 3 blocks of acquisition; see Fig. S1a for SCRs across blocks).

In our second experiment, participants again showed successful acquisition in SCR, exhibiting significantly higher SCRs to the CS+ items than the CS− items (main effect of stimulus, $F(1, 44) = 46.36$, $p < 0.001$, $\eta_p^2 = 0.51$, 95% CI [0.29, 0.65]). Yet in

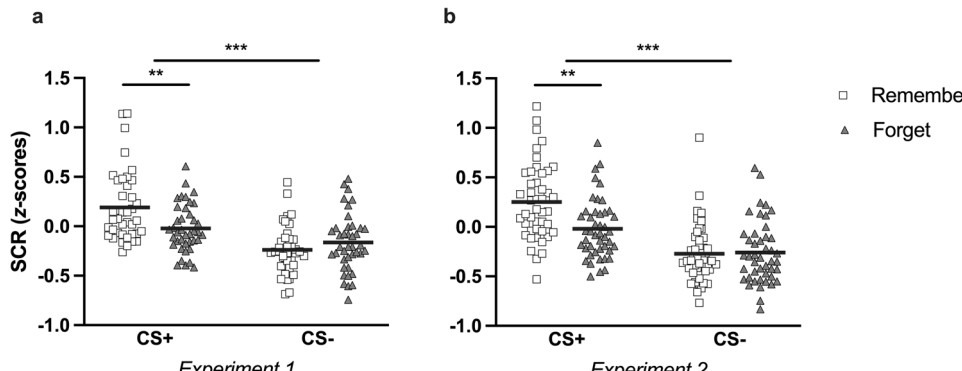

**Fig. 2 Skin conductance responses.** Average SCR (Blocks 2 and 3) per stimulus category, for **a** Experiment 1, and **b** Experiment 2. White squares represent Remember items and gray triangles represent Forget items. Solid black lines depict the group means. **$p < 0.01$, ***$p < 0.001$.

this experiment, higher SCRs were also observed on remember than on forget trials (main effect of instruction, $F(1, 44) = 6.08$, $p = 0.018$, $\eta_p^2 = 0.12$, 95% CI [0.003, 0.30]). Most importantly, the stimulus by instruction interaction was again significant ($F(1, 44) = 8.38$, $p = 0.006$, $\eta_p^2 = 0.16$, 95% CI [0.01, 0.35]) (see Fig. 2b). Replicating the SCR pattern of Experiment 1, follow-up testing indicated evidence for statistically significant differences between remember versus forget items for CS+ ($t(44) = 3.78$, $p = 0.002$, $d = 0.83$, 95% CI [0.21, 1.46]) but not for CS− trials ($t(44) = -0.17$, $p = 1$, $d = -0.04$, 95% CI [−0.63, 0.55]). Of note, some fear learning was obtained in this experiment also under forget instructions, as we observed higher SCR responding for CS + than CS− items not only on remember trials ($t(44) = 7.03$, $p < 0.001$, $d = 1.61$, 95% CI [1.00, 2.22]), but also to some extent on forget trials ($t(44) = 3.22$, $p = 0.011$, $d = 0.74$, 95% CI [0.09, 1.38]). When the factor Block (2,3) was included in the ANOVA, unlike the previous experiment, we obtained evidence for significant differences between our blocks (main effect of block, $F(1, 44) = 13.23$, $p < 0.001$, $\eta_p^2 = 0.23$, 95% CI [0.05, 0.42]) that were influenced by instruction (block by instruction interaction, $F(1, 44) = 9.99$, $p = 0.003$, $\eta_p^2 = 0.19$, 95% CI [0.02, 0.37]). SCRs to remember items decreased from Block 2 to 3 ($t(44) = 4.78$, $p < 0.001$, $d = 0.62$, 95% CI [0.27, 0.97]), while we did not find evidence of a statistically significant difference between Block 2 and Block 3 SCRs to forget items ($t(44) = 0.07$, $p = 1$, $d = 0.01$, 95% CI [−0.34, 0.36]). Significant main effects of stimulus ($F(1, 44) = 46.30$, $p < 0.001$, $\eta_p^2 = 0.51$, 95% CI [0.29, 0.65]) and

instruction ($F(1, 44) = 6.08$, $p = 0.018$, $\eta_p^2 = 0.12$, 95% CI [0.003, 0.30]), in addition to the stimulus by instruction interaction ($F(1, 44) = 8.36$, $p = 0.006$, $\eta_p^2 = 0.16$, 95% CI [0.01, 0.35]) were maintained in this secondary analysis; the triple interaction between stimulus, instruction, and block was non-significant ($F(1, 44) = 1.87$, $p = 0.18$, $\eta_p^2 = 0.04$, 95% CI [0.00, 0.20]). Analyses including all 3 blocks of acquisition can be found in the Supplementary Results and SCRs are depicted in Fig. S1b.

**Free recall.** In terms of item recall performance in Experiment 1, participants recalled a significantly higher proportion of CS+ than CS− items (main effect of stimulus, $F(1, 39) = 7.55$, $p = 0.009$, $\eta_p^2 = 0.16$, 95% CI [0.01, 0.36]), and remember than forget items (main effect of instruction, $F(1, 39) = 39.49$, $p < 0.001$, $\eta_p^2 = 0.50$, 95% CI [0.27, 0.65]) (see Fig. 3a). The interaction was not significant ($F(1, 39) = 1.17$, $p = 0.29$, $\eta_p^2 = 0.03$, 95% CI [0.00, 0.18]). Paired $t$ tests indicated a greater recall of remember than forget items from both the CS+ ($t(39) = 4.66$, $p < 0.001$, $d = 0.94$, 95% CI [0.33, 1.56]) and CS− ($t(39) = 5.85$, $p < 0.001$, $d = 1.18$, 95% CI [0.53, 1.83]) categories.

In examining the correct identification of recalled items as belonging to the CS+ or CS− categories (i.e., associative recall; see Methods for details about the calculation of all outcome measures), we did not detect a statistically significant main effect of stimulus ($F(1, 36) = 3.86$, $p = 0.057$, $\eta_p^2 = 0.10$, 95% CI

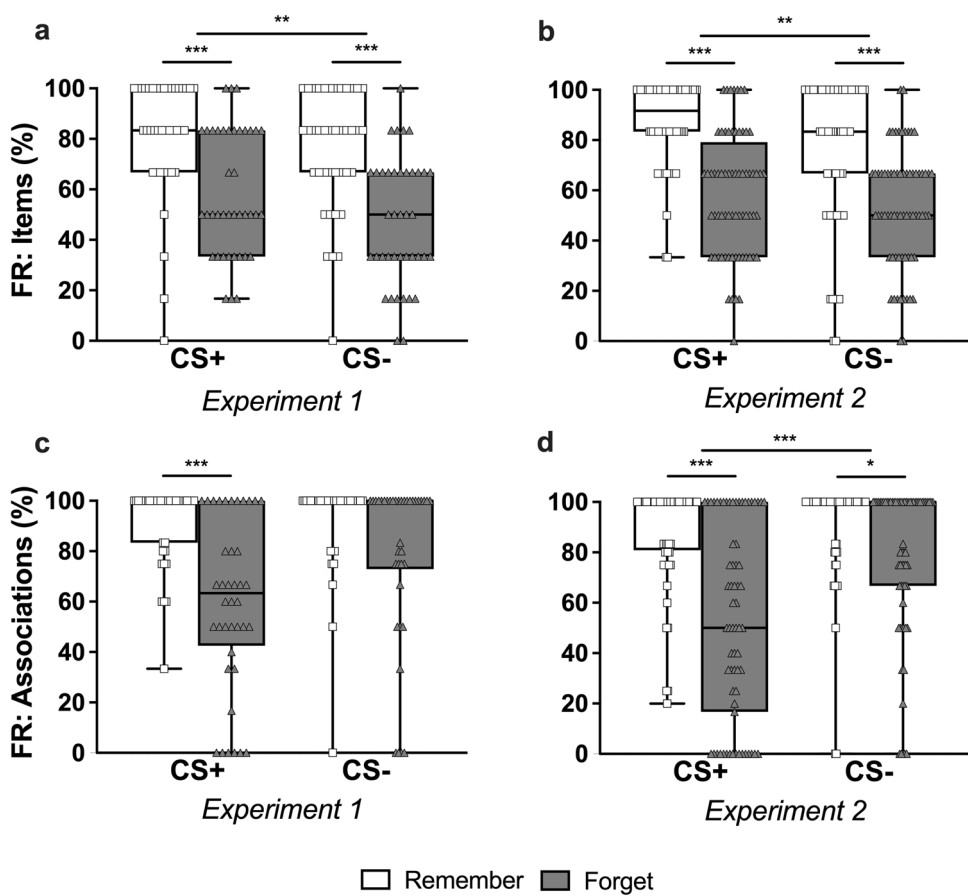

**Fig. 3 Free recall performance.** Free recall performance per stimulus category, for (**a**, **b**) total number of items recalled, irrespective of stimulus (CS+/CS−) categorization errors (expressed as percentage of the total items), and (**c**, **d**) associations correctly identified, for those items correctly recalled (i.e., items correctly categorized/total number of items recalled; expressed as a percentage). Experiment 1 data are displayed in (**a**, **c**) and Experiment 2 data are displayed in (**b**, **d**). White boxes/squares represent Remember items and gray boxes/triangles represent Forget items. Boxes extend from the 25th to 75th percentiles, whiskers extend from the minimum to the maximum, and the center lines depict the medians. *$p < 0.05$, **$p < 0.01$, ***$p < 0.001$.

[0.00, 0.29]). We observed a main effect of instruction ($F(1, 36) = 35.99$, $p < 0.001$, $\eta_p^2 = 0.50$, 95% CI [0.25, 0.65]), with more accurate categorizations of remember than forget items. The interaction between stimulus and instruction did not reach significance ($F(1, 36) = 3.28$, $p = 0.078$, $\eta_p^2 = 0.08$, 95% CI [0.00, 0.28]). Follow-up analyses showed that participants correctly categorized a lower proportion of forget than remember items from the CS+ trials ($t(38) = 5.30$, $p < 0.001$, $d = 1.15$, 95% CI [0.47, 1.84]) but we did not find evidence of a statistically significant difference on the CS− trials ($t(36) = 2.57$, $p = 0.074$, $d = 0.56$, 95% CI [−0.05, 1.16]) (see Fig. 3c).

Participants' total item recall performance in Experiment 2 matched Experiment 1. They recalled significantly more CS+ than CS− items (main effect of stimulus, $F(1, 67) = 8.56$, $p = 0.005$, $\eta_p^2 = 0.11$, 95% CI [0.01, 0.26]), and more remember than forget items (main effect of instruction, $F(1, 67) = 88.73$, $p < 0.001$, $\eta_p^2 = 0.57$, 95% CI [0.41, 0.67]) (see Fig. 3b). The interaction was non-significant ($F(1, 67) = 1.43$, $p = 0.24$, $\eta_p^2 = 0.02$, 95% CI [0.00, 0.13]). Likewise, paired $t$-tests indicated evidence for better recall of remember than forget items for both CS+ ($t(67) = 8.17$, $p < 0.001$, $d = 1.22$, 95% CI [0.74, 1.71]) and CS− trials ($t(67) = 6.70$, $p < 0.001$, $d = 1.00$, 95% CI [0.54, 1.47]).

When participants were asked to indicate whether a recalled item was a CS+ or CS−, we observed main effects of stimulus ($F(1, 63) = 18.63$, $p < 0.001$, $\eta_p^2 = 0.23$, 95% CI [0.07, 0.39]) and instruction ($F(1, 63) = 77.73$, $p < 0.001$, $\eta_p^2 = 0.55$, 95% CI

[0.38, 0.66]), with poorer performance on CS+ items and superior performance on remember items (see Fig. 3d). Unlike Experiment 1, the interaction also reached significance ($F(1, 63) = 6.77$, $p = 0.012$, $\eta_p^2 = 0.10$, 95% CI [0.01, 0.25]). Follow-up testing revealed that participants correctly categorized a higher proportion of remember than forget items from both CS+ ($t(66) = 7.29$, $p < 0.001$, $d = 1.27$, 95% CI [0.72, 1.82]) and CS− trials ($t(64) = 3.08$, $p = 0.015$, $d = 0.54$, 95% CI [0.06, 1.02]), with a larger effect for the CS+ trials.

**Recognition**. In Experiment 1, participants exhibited almost perfect total item recognition (average performance across stimulus categories: 98.33–100%). Accordingly, none of the analyses pointed toward any significant effects (see Fig. 4a). Despite obtaining ceiling effects in total item recognition, we observed the DF effect in the categorization accuracy of recognized stimuli (i.e., associative recognition). Like in free recall, participants were more accurate in categorizing remember than forget items (main effect of instruction, $F(1, 39) = 31.91$, $p < 0.001$, $\eta_p^2 = 0.45$, 95% CI [0.21, 0.61]). We did not observe statistically significant differences in performance between CS+ and CS− items ($F(1, 39) = 0.05$, $p = 0.83$, $\eta_p^2 = 0.001$, 95% CI [0.00, 0.05]) or an interaction with instruction ($F(1, 39) = 0.11$, $p = 0.74$, $\eta_p^2 = 0.003$, 95% CI [0.00, 0.10]). Planned $t$-tests pointed to more precise categorization of remember than forget items from CS+

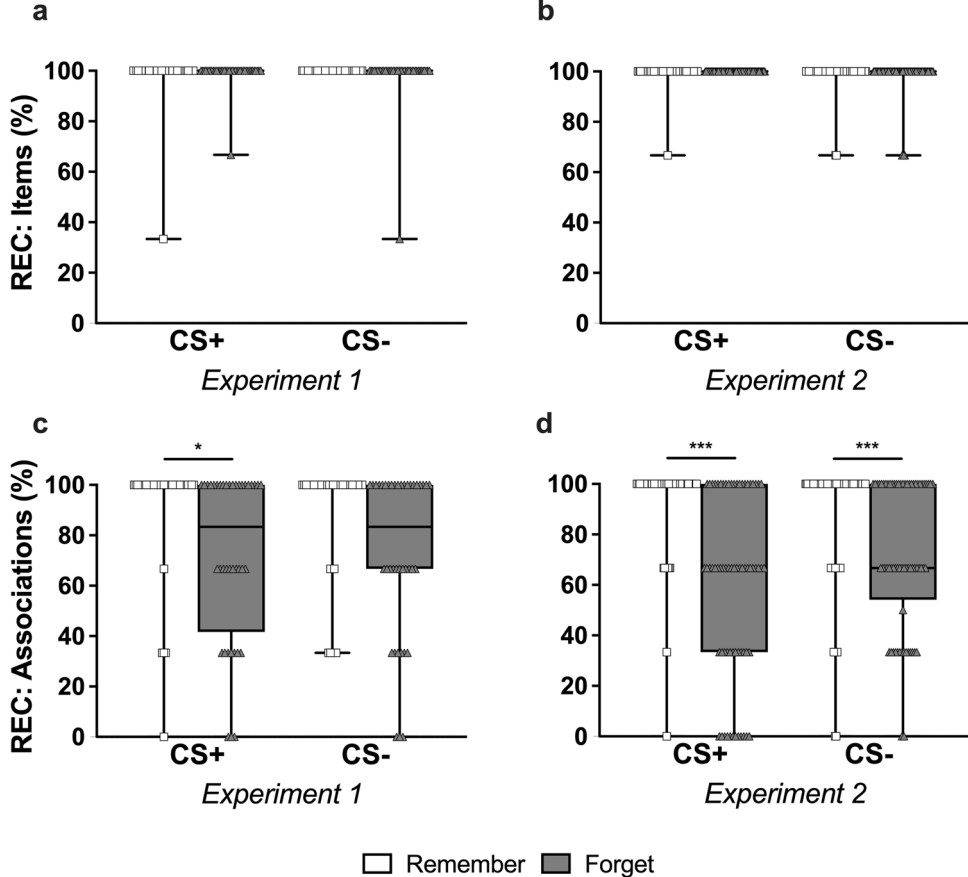

**Fig. 4 Recognition performance.** Recognition performance per stimulus category, for (**a**, **b**) total number of items recognized, irrespective of stimulus (CS+/CS−) categorization errors (expressed as percentage of the total items), and (**c**, **d**) associations correctly identified, for those items correctly recognized (i.e., items correctly categorized/total number of items recognized; expressed as a percentage). Experiment 1 data are displayed in (**a**, **c**) and Experiment 2 data are displayed in (**b**, **d**). White boxes/squares represent Remember items and gray boxes/triangles represent Forget items. Boxes extend from the 25th to 75th percentiles, whiskers extend from the minimum to the maximum, and the center lines depict the medians. *$p < 0.05$, ***$p < 0.001$.

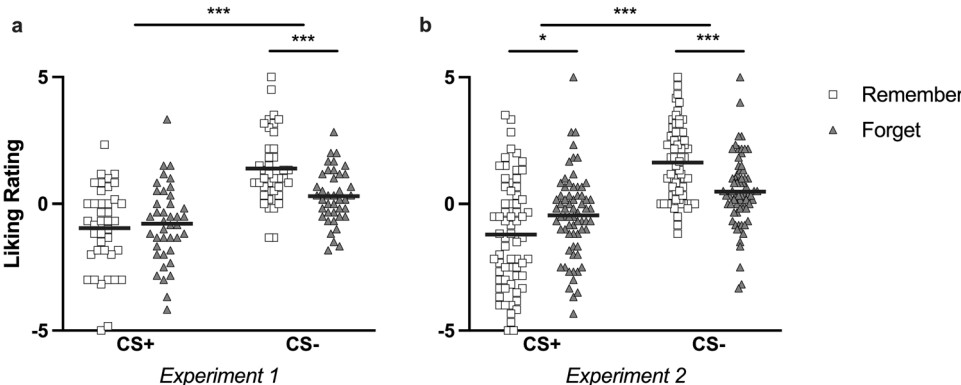

**Fig. 5 Liking ratings.** Average liking ratings per stimulus category, for **a** Experiment 1, and **b** Experiment 2. White squares represent Remember items and gray triangles represent Forget items. Solid black lines depict the group means. *$p < 0.05$, ***$p < 0.001$.

trials ($t(39) = 3.25$, $p = 0.011$, $d = 0.68$, 95% CI [0.08, 1.27]), but we did not find evidence of a statistically significant difference on the CS− trials ($t(39) = 2.68$, $p = 0.055$, $d = 0.56$, 95% CI [−0.02, 1.14]) (see Fig. 4c).

Similarly, in Experiment 2, participants again exhibited almost perfect total item recognition (average performance across stimulus categories: 98.04–100%), and once more, all comparisons were non-significant (see Fig. 4b). Despite the lack of differences in total item recognition, again, stimulus categorization accuracy differed as a function of instruction. As expected, participants were significantly more accurate in categorizing remember than forget items (main effect of instruction, $F(1, 67) = 91.72$, $p < 0.001$, $\eta_p^2 = 0.58$, 95% CI [0.42, 0.68]). We found no evidence for a statistically significant difference on performance between CS+ and CS− items (main effect of stimulus, $F(1, 67) = 1.06$, $p = 0.31$, $\eta_p^2 = 0.02$, 95% CI [0.00, 0.11]), yet in this experiment, the interaction did reach significance ($F(1, 67) = 7.74$, $p = 0.007$, $\eta_p^2 = 0.10$, 95% CI [0.01, 0.25]). Like in the free recall task, participants correctly categorized a higher proportion of remember items from both CS+ ($t(67) = 8.29$, $p < 0.001$, $d = 1.27$, 95% CI [0.77, 1.78]) and CS− trials ($t(67) = 4.03$, $p < 0.001$, $d = 0.62$, 95% CI [0.19, 1.05]), with a slightly larger effect for CS+ trials (see Fig. 4d).

**Liking ratings**. Analysis of liking ratings revealed a significant interaction between stimulus and instruction ($F(1, 39) = 16.61$, $p < 0.001$, $\eta_p^2 = 0.30$, 95% CI [0.08, 0.48]) in Experiment 1. Ratings were generally higher for remember than forget associations (main effect of instruction, $F(1, 39) = 9.38$, $p = 0.004$, $\eta_p^2 = 0.19$, 95% CI [0.02, 0.39]) and in line with the association of CS+ items with an unpleasant US, CS− items received higher liking ratings than CS+ items (main effects of stimulus, $F(1, 39) = 26.02$, $p < 0.001$, $\eta_p^2 = 0.40$, 95% CI [0.16, 0.57]) (see Fig. 5a). We found no evidence for a statistically significant difference between ratings of remember and forget items from CS+ trials ($t(39) = −0.81$, $p = 1$, $d = −0.12$, 95% CI [−0.53, 0.28]), but participants reported higher liking of remember items than forget items from CS− trials ($t(39) = 5.06$, $p < 0.001$, $d = 0.76$, 95% CI [0.30, 1.23]).

The significant stimulus by instruction interaction in the liking ratings was maintained in Experiment 2 ($F(1, 67) = 34.41$, $p < 0.001$, $\eta_p^2 = 0.34$, 95% CI [0.16, 0.48]). Participants reported to like CS− items more than CS+ items (main effect of stimulus, $F(1, 67) = 56.89$, $p < 0.001$, $\eta_p^2 = 0.46$, 95% CI [0.28, 0.58]), but we found no statistically significant difference between remember and forget items (main effect of instruction, $F(1, 67) = 1.28$, $p = 0.26$, $\eta_p^2 = 0.02$, 95% CI [0.00, 0.12]). In follow-up testing, we observed a higher rating for remember than forget items from CS− trials ($t(67) = 4.80$, $p < 0.001$, $d = 0.68$, 95% CI [0.27, 1.09]),

and a lower rating for remember than forget items from CS+ trials ($t(67) = -3.13$, $p = 0.013$, $d = -0.45$, 95% CI [−0.84, −0.05]) (see Fig. 5b).

## Discussion

In two experiments, we set out to investigate whether DF can effectively be applied to disrupt the encoding of emotional associative memories. Across experiments, participants reliably exhibited reduced item/associative recall and reduced associative recognition memory for information they had been instructed to forget during encoding. We thus obtained robust evidence for diminished declarative memory expression for associative information as a result of our DF manipulation. Moreover, we observed an attenuation of the physiological expression of fear memory (SCR responding) for shock associations that participants were instructed to forget. Taken together, our results show that the encoding of emotional associative memory can be disrupted through DF.

While findings were generally consistent between experiments, a few minor inconsistencies were observed and deserve comment. During the recognition task, and across both experiments, we observed identical and almost perfect total item recognition rates for all stimulus categories (i.e., all follow-up comparisons between CS+ remember/forget and CS− remember/forget were non-significant). A clear DF effect was however detected in the accuracy with which those stimuli were categorized as CS+ or CS− (i.e., associative recognition), be it that the precise expression of the effect differed slightly between experiments. In Experiment 1, participants were less accurate in categorizing CS + F items than CS + R items, but there was no evidence for a statistically significant difference in their performance on CS− items (see Fig. 4c). However, in Experiment 2, participants were less accurate in categorizing forget than remember items from both CS+ and CS− trials (i.e., a global DF effect; see Fig. 4d). Nonetheless, our preregistered follow-up analyses revealed that across experiments, we observed a larger DF effect on CS+ trials (i.e., comparison of CS+R versus CS+F) than CS− trials (i.e., comparison of CS−R versus CS−F). Additionally, our DF manipulation had a stronger effect on CS+ associative recognition performance in Experiment 2 than in Experiment 1 ($d = 1.27$ versus $d = 0.68$, respectively). We can only speculate as to the slight differences observed between experiments in associative recognition performance, given that the DF procedure in both experiments was identical. Perhaps these discrepancies reflect differences in obtained power given the $N$ of each experiment. Alternatively, they may be attributable to sample variations, as the experiments were conducted about 2 years apart.

In the free recall task, we found a global DF effect for the total items recalled (not accounting for errors in stimulus

categorization) that was very similar across both experiments. Participants recalled a smaller proportion of items from trials they had been instructed to forget, whether they were CS+ or CS− items (see Fig. 3a, b). When categorizing the recalled stimuli as CS+ or CS− (i.e., associative recall), participants in Experiment 1 were less accurate in categorizing CS+F than CS+R items specifically (see Fig. 3c), whereas we observed a global DF effect in Experiment 2, with participants being less accurate in categorizing forget than remember trials in general (see Fig. 3d). Again, we consistently detected a larger DF effect in associative recall for the CS+ than the CS− trials in both experiments, but unlike for the recognition task, the disruption of CS+ associative recall due to DF was comparable between the two experiments (effect sizes of $d = 1.15$ and $d = 1.27$, respectively). Thus, when examining associative rather than item memory in free recall, we observed a very similar pattern of results as in the recognition task.

While DF research typically involves a single presentation of items, in our experiments we presented all items and their associated outcomes three times, to be able to track the development of differential conditioned fear responding (SCR) to the CS+ and CS− items and its attenuation through DF. Our procedure was successful in evoking differential SCRs and crucially, our DF intervention was effective at disrupting physiological fear memory expression, through mere instruction. It should be noted that some fear conditioning was still maintained under forgetting instructions in Experiment 2, but this is hardly surprising given the robust, immediate learning that transpires during fear conditioning; if anything, it is remarkable that directed forgetting occurred at all during our procedure as the forgetting instructions are working in opposition here to the fact that the repeated pairings of the CSs invite remembering rather than forgetting. Regarding the nature of the associative memories being created with the current design, we have to remain agnostic as to whether participants rely on separate episodic memories for each pairing of a given CS with the presence or absence of the US or whether they rely on one representation that is updated or strengthened with each repetition. Both of those possibilities would be compatible with our results; the directed forgetting manipulation could interfere with the formation of specific episodic memories or with the acquisition and strengthening of a non-episodic memory representation.

We did notice some differences between declarative and physiological indices of memory expression in our experiments. In declarative report, the item recall data support a global DF effect, as participants exhibited a clear memory deficit in recalling to-be-forgotten items, whether they were CS+ or CS−. Yet, upon subsequent stimulus categorization, which reflects the memory of the association of each item with either the US or its absence, both recognition and recall data reflect an enhanced forgetting effect for CS+ items that was maintained across both experiments. This observation parallels the SCR findings, where we also obtained a DF effect for CS+ items only. Robust effects on CS+ performance across three separate outcome measures allow us to suggest that DF may be a suitable manipulation to interfere with the encoding of emotional associations. Importantly, while one could argue that declarative outputs may be sensitive to demand effects (even though this idea has been refuted as a major source of directed forgetting effects in prior experiments[35,36]), compliance could not possibly account for our SCR findings, given that SCR reflects activity of the autonomic nervous system, and thus, its expression is impervious to participants' control.

Enhanced effects for the CS+ were not seen in the liking ratings, where participants reported a similar dislike for CS+R and CS+F items in Experiment 1, and a slightly stronger dislike for CS+R than CS+F items in Experiment 2 (but see Supplemental Results for additional analyses using a sample of $N = 45$ for

Experiment 2). Unlike all our other measures, in the liking ratings, we obtained stronger effects on CS− trials, as participants reported reliably stronger liking for CS−R than CS−F items. It is conceivable that the liking ratings, which were obtained at the end of the experiment, were tainted by the multiple prior exposures to all CS stimuli (e.g., three times during acquisition, then during free recall and recognition, and then once again during the rating task). In future studies, it would be helpful to obtain US expectancy ratings during the acquisition phase, as to be able to compare controlled and non-controlled readouts of memory more directly.

Our findings are in line with decades worth of prior DF research using neutral stimuli[37], but they are in contrast with the results of studies that tested non-associative negative emotional stimuli[21–23]. Such studies reported higher recall/recognition of negative stimuli, which, consequently, yielded diminished DF effects (but see[38] for an investigation of different categories of emotional stimuli). Further, a recent meta-analysis of item-method DF confirmed that emotional memories are often more resilient to forgetting than neutral memories, but several moderating factors were also reported[39]. In our data, negative valence did not weaken the DF effect, as is reflected in the robust disruption of CS+ retention in both verbal and non-verbal indices of fear memory. Thus, with the current design, we demonstrate that it is indeed possible to interfere with memory for emotional information through DF and bring additional evidence to the field.

**Limitations**. Along with the evidence obtained, some limitations of our experiments should be noted. Unlike other item-method DF studies, our procedure involved the repeated presentation of all items, with half of those items being paired with aversive electrical stimulation. In order to accomplish this without administering an excessive number of shocks to our participants (which arguably would lead to their habituation to shock), we had to reduce the number of CS stimuli to 24, which is less than commonly used in item-method DF procedures and could have led to the ceiling effects observed in item recognition (even though a similarly small number of stimuli has been used in list-method DF; see e.g., ref. [40]). Further, other DF research has employed working memory tasks in between the DF manipulation and follow-up tests, or even during ITIs, as to prevent rehearsal and purge working memory before testing. We did not do so in these experiments, as we wanted to first establish whether we could observe a DF effect for emotional associative memories to begin with, but such tasks will be useful in future research as we try to elucidate the underlying mechanisms responsible for the observation of DF effects in this specific procedure. Most importantly, with this procedure we could not directly compare the effects obtained during acquisition (i.e., in SCR) and during retention testing (i.e., in recognition and free recall), which may or may not reflect differences between learning and memory, respectively. While it would be difficult to design a study where acquisition data is compared to free recall data (given that some items and their associations will not be recalled at all, making it a biased comparison), it would be possible to compare acquisition data to recognition data if all acquisition stimuli are included in the recognition task (which was not the case for the experiments presented here).

**Potential underlying mechanisms**. While the current set of studies provides evidence for DF of emotional associative memories, these studies were not designed to allow unequivocal inferences about potential underlying mechanisms. Yet in Experiment 2, we tested certain individual difference factors that we believed were relevant for shedding light on fundamental DF mechanisms.

Despite failing to obtain insightful results from those analyses (see Supplementary Results and Table S1 for details on the individual differences analyses), we can still speculate about possible mechanisms based on the rich literature on DF for neutral and non-associative memories. In that literature, item-method DF effects have often been attributed to selective rehearsal, a mechanism that implies a disruption during the encoding phase. According to this account, items that are presented one by one are held in working memory until the instruction to remember or forget is introduced; participants are then thought to continue actively rehearsing the R items, while stopping to rehearse F items, which eventually passively decay[41–43]. It might be difficult to maintain this explanation for the findings of our experiments, where all items were repeated, and thus, additional rehearsal of both R and F items should have occurred (due to mere repetition), be it perhaps to a lesser extent for F items than for R items (due to the F cues halting rehearsal). However, we observed ceiling performance in item recognition across experiments, which suggests that rehearsal was indeed actively taking place, but it was not selective to remember items only. Nonetheless, future studies could introduce variable delays between CS presentations and instruction cues or a concurrent taxing working memory task to see if disrupting rehearsal can diminish the DF effect, in support of the selective rehearsal hypothesis.

Another candidate for explaining DF effects, that has been equally advocated in the literature, is retrieval inhibition. This account starts from the observation that in DF procedures, stimuli are typically presented in isolation at trial onset, and only then followed by the instruction cue, hence, F items are proposed to first be encoded, but later inhibited during retrieval (i.e., disruption during retrieval)[9]. However, re-presenting the stimuli later, such as in a recognition task, should serve to reverse such inhibition, as the stimuli then act as retrieval cues, which should produce similar recognition of R and F items on a later test[10]. According to this logic, retrieval inhibition could explain the results of our almost-perfect item recognition performance in both experiments. However, if this were the case, we should not have observed a DF effect at all on the associative recognition performance. As such, retrieval inhibition does not seem a plausible mechanism to explain our results either.

A third popular account, attentional inhibition, proposes that upon presentation of an F instruction, attentional mechanisms are engaged that actively suppress the processing of information that has become goal-irrelevant (i.e., F items). Attentional inhibition has also been suggested to be responsible for suppressing any goal-irrelevant information that may have entered working memory and for preventing individuals' attention from returning to information that was previously deemed goal-irrelevant (e.g., CS stimuli following an F instruction in our case)[11]. In line with this idea, neural evidence from the memory suppression literature, as well as from item-method DF, favors an active inhibitory process during the encoding phase[12,44–47] (for a discussion, see ref. [48]), as functional imaging studies have found increased right prefrontal activation, as well as reduced hippocampal activity during to-be-forgotten trials[12,44,45,49]. This would suggest then that engaging prefrontal inhibitory processes during the encoding phase to actively suppress F items may prevent later memory retrieval (i.e., disruption during encoding)[50]. This proposal can be a candidate explanation for our results, but again, further research is necessary to examine if the differences in neural activations are maintained following repeated presentations of the same stimuli in combination with recurring R/F instructions as in our procedure, which may arguably reduce goal-irrelevance for the F items.

In search of an explanation for our findings, one may find inspiration in the work of Anderson and colleagues on retrieval suppression using the Think/No-Think (TNT) paradigm[51,52]. Extensive research on retrieval suppression has recently led to the proposal of the retrieval stopping model[53] of fear extinction, which can be extended to explain our results here. According to this model, when a person encounters a reminder of an unwanted memory, they can terminate (or suppress) the retrieval of that memory by engaging prefrontal cortical regions (most notably the right dorsolateral prefrontal cortex; rDLPFC) that down-regulate the amygdala and suppress further processing in the hippocampus. Notably, retrieval suppression has been shown to affect both subjective and psychophysiological (e.g., SCR, heart rate deceleration) indices of emotion, and it would allow for a straightforward explanation of our results[54,55]. Indeed, this model may very well account for the results that we obtained in free recall and recognition, as well as explain why our participants were able to stop the retrieval of the (unwanted) emotional associations. As for the SCR results, one may argue that from the second presentation of each CS onwards, retrieval suppression can prevent the retrieval of the association established through the first CS presentation, thereby impairing SCR expression. As such, this model could provide a plausible explanation of our results. Whether it indeed does, remains however to be tested in future research addressing the neural underpinnings of the current findings.

## Conclusions

Thus, while most explanations for DF effects focus on a disruption of encoding, the work of Anderson and colleagues[48,56] on memory suppression in humans inspires the idea of a combination of inhibitory processes that are active at encoding and retrieval, working jointly to suppress unnecessary or unwanted memories. Our data, like other published DF data, indicate that DF is a graded phenomenon: at least some F associations were encoded, as recognition performance for F associations was above chance in both experiments. Thus, in future research, it would be interesting, also from a clinical perspective, to focus on boosting disruption of retrieval, as humans often seek to forget aversive experiences that have long since been encoded and consolidated. Along those lines, studies using memory suppression techniques have already provided some primary evidence demonstrating reduced recall of unwanted memories following subliminal memory reactivation[57] and reduced (self-reported) symptoms of anxiety, negative affect, and depression, following a 3-day, online suppression training[58]. Even though extensive further research is required before advancing to clinical translation, our data across a set of experiments convincingly show that DF manipulations can be successfully used to interfere with emotional memory expression. In combination with the fact that more than half a century's worth of DF research has repeatedly yielded strong effects that are robust across different memory domains and procedures, it appears that instructed forgetting is a procedure that holds promise for the creation of novel emotional memory modification protocols in the future.

## Data availability

The final datasets generated and used for analyses and all analyses outputs are publicly available on the Open Science Framework (OSF) at https://osf.io/65x7a/.

## Code availability

The analysis code and experimental code can be publicly accessed at https://osf.io/65x7a/.

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

## Acknowledgements

This work was funded by Consolidator Grant 648176 of the European Research Council (ERC) awarded to Tom Beckers. Anastasia Chalkia was supported by a personal fellowship awarded by the KU Leuven Research Council (PDM/20/061). This research was also supported by an infrastructure grant from the Research Foundation Flanders (FWO) and the Research Fund of KU Leuven, Belgium (I011320N; AKUL/19/06). The funders had no role in the study design, data collection and analysis, decision to publish or preparation of the manuscript. The authors would like to thank Mathijs Franssen and Jeroen Clarysse for their essential technical support, as well as Liesbeth Carpentier and Jade Willaert for their assistance with the data collection.

## Author contributions

A.C., N.V., L.V.O., M.K., and T.B. contributed to the designs of the experiments. A.C. and N.V. were involved in the execution of the experiments and the data collection. A.C. and N.V. processed and analyzed the data. A.C., N.V., L.V.O., M.K., and T.B. interpreted the data. A.C. drafted the original manuscript and all authors critically reviewed, revised, and approved the final manuscript.

## Competing interests

The authors declare no competing interests.
