## [Peer Review File · Communications Psychology]

17th May 23

Dear Dr Chalkia,

Thank you for your patience during the peer-review process. Your manuscript titled "Disrupting emotional associative memory through directed forgetting" has now been seen by 3 reviewers, and I include their comments at the end of this message. They find your work of interest, but raised some important points. We are interested in the possibility of publishing your study in Communications Psychology, but would like to consider your responses to these concerns and assess a revised manuscript before we make a final decision on publication.

We therefore invite you to revise and resubmit your manuscript, along with a point-by-point response to the reviewers. Please highlight all changes in the manuscript text file.

All reviewers provided a positive and constructive assessment of your paper. We ask you to revise the manuscript and include the analysis requested by Reviewer 2. Please ensure you label this as analysis as not pre-registered. Reviewer #1 (and also Reviewer #2) raise a list of presentational issues that we ask you to address. In addition to requested changes to other parts of the text, please ensure that you include a section in the Discussion under the sub-heading "Limitations" to discuss any limitations and mention important caveats. Reviewer 3 suggests discussion of additional references. These are no doubt valuable suggestions. Please be advised that we expect a fair and comprehensive treatment of existing work, but not an exhaustive review of the literature.

Finally, Reviewer 2 mentions some concerns regarding the availability of Code. As detailed below, the journal requests the release of custom analysis code (and experimental code where appropriate). In case of publication, you will also be requested to publicly deposit the numerical data underlying graphs and figures (and encouraged to also share the full data in anonymized form if possible). To avoid future delays, we recommend preparing these depositions already. More information can be found below.

Please use the following link to submit your revised manuscript, point-by-point response to the referees' comments (which should be in a separate document to any cover letter) and the completed checklist:
[link redacted]

We hope to receive your revised paper within 4 weeks; please let us know if you aren't able to submit it within this time so that we can discuss how best to proceed. If we don't hear from you, and the revision process takes significantly longer, we may close your file. In this event, we will still be happy to reconsider your paper at a later date, provided it still presents a significant contribution to the literature at that stage.

Please do not hesitate to contact me if you have any questions or would like to discuss these revisions further. We look forward to seeing the revised manuscript and thank you for the opportunity to review your work.

Best regards,

Antonia Eisenkoeck

Antonia Eisenkoeck
Senior Editor
Communications Psychology

EDITORIAL POLICIES AND FORMATTING

Editorial Policy: [Policy requirements](https://www.nature.com/documents/nr-editorial-policy-checklist.pdf) (Download the link to your computer as a PDF.)

Furthermore, please align your manuscript with our format requirements, which are summarized on the following checklist:

[Communications Psychology formatting checklist](https://www.nature.com/documents/commspsychol-style-formatting-checklist-article-rr.pdf)

and also in our style and formatting guide [Communications Psychology formatting guide](https://www.nature.com/documents/commspsychol-style-formatting-guide-accept.pdf) .

* **CODE AVAILABILITY:** All Communications Psychology manuscripts must include a section titled "Code Availability" at the end of the methods section. In the event of publication, we require that the custom analysis code supporting your conclusions is made available in a publicly accessible repository; at publication, we ask you to choose a repository that provides a DOI for the code; the

link to the repository and the DOI will need to be included in the Code Availability statement. Publication as Supplementary Information will not suffice. We ask you to prepare code at this stage, to avoid delays later on in the process.

*** DATA AVAILABILITY:**

All Communications Psychology manuscripts must include a section titled "Data Availability" at the end of the Methods section or main text (if no Methods). More information on this policy, is available at <http://www.nature.com/authors/policies/data/data-availability-statements-data-citations.pdf>.

At a minimum the Data availability statement must explain how the data can be obtained and whether there are any restrictions on data sharing. Communications Psychology strongly endorses open sharing of data. If you do make your data openly available, please include in the statement:

We recommend submitting the data to discipline-specific, community-recognized repositories, where possible and a list of recommended repositories is provided at <http://www.nature.com/sdata/policies/repositories>.

If a community resource is unavailable, data can be submitted to generalist repositories such as [figshare](https://figshare.com/) or [Dryad Digital Repository](http://datadryad.org/). Please provide a unique identifier for the data (for example a DOI or a permanent URL) in the data availability statement, if possible. If the repository does not provide identifiers, we encourage authors to supply the search terms that will return the data. For data that have been obtained from publicly available sources, please provide a URL and the specific data product name in the data availability statement. Data with a DOI should be further cited in the methods reference section.

REVIEWERS' EXPERTISE:

Reviewer #1: directed forgetting, associative memory

Reviewer #2: directed forgetting, fear conditioning

Reviewer #3: directed forgetting

REVIEWERS' COMMENTS:

Reviewer #1 (Remarks to the Author):

This paper addresses an interesting question, which is whether a directed forgetting instruction at the time of encoding reduces memory for emotionally significant associations, in particular as measured with an objective skin conductance measure but also with a number of self-report measures. At face value, the results provide suggestive evidence for directed forgetting on all measures, at least for the CS+ stimulus that was paired with electric shock.

My main comments relate to the theoretical interpretation of the results and the conclusions that can be drawn from them.

First, the authors conclude that their results show that forgetting instructions can interfere with emotional associative memory (last sentence of Abstract). However there was no manipulation of the emotional content of the outcome. Of course the use of an innocuous outcome would preclude measurement of skin conductance responses, but it would still be informative to directly test the impact of emotional content on the self-report memory measures. If there was no effect of emotional content on these measures, it would suggest that there is nothing special about emotional memories and that skin conductance might simply be another readout of a memory when the content happens to be emotional.

Related to the above point, the authors motivate their study by referring to a possible dissociation between “declarative” memory and “emotional” memory, without making it clear what they mean by this distinction outside the narrow field of directed forgetting. In the Discussion, they fail to acknowledge that their different measures were broadly consistent, thus allowing the possibility of a single memory system regardless of content.

The authors measured both memory for specific CS items as well as for CS-US mappings, and they switch back and forth between the results so that it is sometimes unclear which one they are referring to (e.g. in the discussion of retrieval inhibition on page 18). The question about how these two types of memory relate to each other is an important one. In the context of an associative learning paradigm with multiple presentations of each pairing (3 in this experiment), are we to assume participants have separate episodic memories for each pairing, or do they form a general representation of the pairing on the first trial which is then augmented on subsequent trials? More generally, is there a distinction between learning and memory? It would be good for the authors to address these issues and explain to what extent a memory framework is appropriate to describe what happens in associative learning.

In the Discussion, I think the authors dismiss the selective rehearsal explanation of their results too readily (top of page 18). The use of multiple trials does not reduce the relevance of rehearsal of CS-US pairs. A rehearsal explanation fits with the procedure of the study more naturally than retrospective inhibition or interference with an established associative memory.

Conversely, the logic of attentional inhibition in terms of freeing up cognitive resources (pages 18-19) does not apply readily to the present design, where forget and control pairings were spaced apart by an ITI of 15-10s and hence did not directly compete for processing.

In conclusion, I think this paper is publishable as preliminary evidence for directed forgetting of associative memory, but it would benefit from some rewriting to acknowledge the theoretical issues and limitations I have noted above.

Detailed comments

- 1) In Figure 1 shock should be explicitly represented in the diagram for CS+ trials
- 2) When was the working memory task given?
- 3) Page 25: the instructions given to participants presumably referred to “shock” rather than “US”?

Reviewer #2 (Remarks to the Author):

Summary: In two experiments, the authors combined a fear conditioning procedure with the item-method of directed forgetting and assessed recall and recognition of CSs as well as CS-US associations, CS liking, and skin conductance responses. They found that to-be-forgotten CSs and CS-US associations were recalled and recognized less than to-be-remembered CSs and CS-US associations. Furthermore, importantly, participants' skin conductance response was reduced for threat-associated CS+ stimuli when paired with forget instructions.

Evaluation: The manuscript addresses a highly relevant and timely research question. It is well written and the results and corresponding conclusions to be drawn are clear. I recommend that the manuscript is accepted after a minor revision and detail my comments below.

Minor:

1. On page 4, the authors list stimulus-response (S-R) associations as declarative associative memories. However, stimulus-response associations are most often argued to be procedural in nature. Furthermore, recent findings raise some doubt as to whether such procedural stimulus-response associations can be forgotten based on instructions, that is, whether their behavioral effects, like repetition priming effects, can be affected by forget instructions (e.g., Dames et al., 2022, <https://escholarship.org/uc/item/1s48z1z6>). Of course, it is reasonable to assume that there is also a declarative stimulus-response association that is, for instance, the basis of explicit knowledge regarding the stimulus-response association and can be assessed via recall or recognition performance (rather than behavioral effects). Nevertheless, I suggest that the authors update their corresponding description to better accommodate perspectives on stimulus-response associations that focus on their procedural aspects by, for instance, clarifying that they merely refer to aspects related to the recall/recognition of experienced stimulus-response pairings.
2. As far as I could see, the authors did not explicitly test whether fear conditioning effects still persisted after a forget instruction (i.e., comparison of CS+F and CS-F). I believe that information on these test should be added, as whether forgetting fully abolishes fear conditioning effects or merely reduces them is of theoretical importance and a relevant piece of information when it comes to the discussion of underlying mechanisms. Based on the figures, it appears as if fear conditioning effects persist even after forgetting at least in Experiment 2. Correspondingly, I would like to recommend that the authors also discuss this finding specifically. That is, in the authors' opinion, can any of the assumed mechanisms be excluded based on the finding that fear conditioning persists even after forgetting?

3. The authors' finding that a higher proportion of associations with CS+ as compared to CS- were forgotten seems to contradict the notion that emotionally-relevant content is typically remembered better/forgotten less readily. In their discussion, could the authors elaborate a bit more on their thoughts regarding their finding that association recall was worse for CS+ as compared to CS- after forget instructions?
4. Other studies assessing directed forgetting most often use some task (e.g., Corsi task) to purge working memory before testing recall/recognition. Was there a particular reason why this was not done in the present experiments?
5. I especially appreciate that the authors differentiated between forgetting of the stimulus and forgetting of the association. This aspect could be stated a bit more explicitly in the introduction to further guide the readers and prevent misconceptions that are only resolved later.
6. The authors have not shared their analysis scripts and have used several programs for analysis. I would appreciate it if the authors could briefly justify why different analysis programs had to be used and why they did not share their analysis scripts. It would be ideal if they could also share their analysis scripts.

Reviewer #3 (Remarks to the Author):

Review of COMMSPSYCHOL-23-0077-T

Disrupting emotional associative memory through directed forgetting

Journal: Communications Psychology

Authors: Chalkia, Vanhasbroeck, Van Oudenhove, Kindt, & Bekers

Summary. The impacts of directed forgetting instructions on both declarative memory and conditioned threat responses were examined in two experiments that integrated the classic item-method directed forgetting procedure and fear conditioning. Twenty-four line drawings were presented, one at a time for 8 seconds apiece, and participants were directed to memorize these drawings for a later test. Half of these pictures ended their 8 second presentation with a brief electric shock (CS+) and half did not (CS-). It was participants' understanding that they were to memorize which pictures went with shock and which did not. Shortly following the shock, for half of the CS+ and half of the CS- items, participants received a tone indicating that they could forget the preceding item. In an unusual feature of the current design, the 24 stimuli were repeated over 3 cycles, in each case with a consistent remember or forget instruction and a consistent shock/non-shock. This type of repetition never happens in item method directed forgetting procedures but was necessitated by the need to measure whether the shocks during round 1 had indeed triggered an association between the picture and the shock, which is only measurable on rounds 2 and 3. After the 3 cycles, participants free recalled all the pictures and tried to label which pictures had shocks. They then did recognition tests on the pictures—both an item recognition test (do you recognize this picture) and an associative recognition test (for each picture, choose whether it was shocked or not). Finally liking ratings were done on all pictures. The two experiments were largely identical except that the second one was done 2 years later, was pre-registered, and included a boatload of executive function and individual differences measures which did not yield interesting insights. The authors found significant directed forgetting effects in free recall and recognition, extending past work in a repeated exposures paradigm. Critically, they also found that directed forgetting affected electrophysiological indices of fear (SCR), reducing them for to-be-forgotten items. Interestingly this affective forgetting effect was selective for the CS+ items, and directed forgetting effects in

associative recall and recognition also showed larger effects for CS+ than CS-. The broad theme here is that—in contrast to directed forgetting studies that focus on affective pictures or words, directed forgetting was found to be greater for fearful content than for neutral content on measures that would potentially be sensitive to the fear association (i.e. conditioning; but also associative recall and recognition; but not item recognition or free recall).

Comments. It was a genuine pleasure to read this paper. I was seriously impressed not only by the quality of the scientific effort, but also by the highly consistent and persuasive findings in SCR and associative measures, greater for CS+ than CS-. It was also made more convincing by the preregistration of the effects. This is certainly the kind of noteworthy paper that should be published in *Communications Psychology*.

And in a highly unusual turn of events, I actually have little to say by way of critical commentary of the work itself, or its interpretation. Statistics are largely appropriate; questions are important and highly relevant to readership of this journal. This was simply a very well-done study—very professional, very careful, and supremely interesting. Well done.

On the other hand, I have a bit more to say about the scholarship of the paper. I was struck by the unfortunately narrow focus of the paper on research using the directed forgetting paradigm, with a virtual absence of highly relevant work on memory control derived from the Think/No-Think procedure. The readership would clearly benefit from a synthesis of these bodies of work in this paper and this would need to be done to integrate the finding with the field.

Here are some notable points that I would like the authors to address to relate this work on item method directed forgetting to retrieval suppression in the Think/No-Think task:

1. Can Memory Control Processes Disrupt Affective Representations? One might get the sense that this has never been done before from the paper, and this actually is largely true within the confines of the item-method directed forgetting procedure, at least when it comes to (a) psychophysiological indices of emotion and (b) conditioning. However, in work on retrieval suppression (where people are given a cue to a previously associated memory and asked to stop retrieval), it has now been repeatedly found that retrieval suppression affects both subjective and psychophysiological indices of emotion (e.g., Gagnepain et al. 2017, *Jneuro*; Legrand et al. Scientific reports; Harrington et al. *Clinical Psychological Science*; Nishiyama & Saito, 2022; Mamat & Anderson 2023, *PsyRxiv*) and these effects have been tied to regulation of the amygdala. The latter fact seems like it would be helpful in putting a mechanistic face on why the current effects may impact both SCR and associative components relating to fear.

2. What might the relationship between fear conditioning and intentional forgetting be? The authors may be interested to learn of our recent attempt to synthesize work on motivated forgetting and fear conditioning. We have a paper in *Neuropsychopharmacology* with Stan Floresco synthesizing rodent and human literature on this, and have offered the “Retrieval stopping model of fear extinction” in which we posit that fear extinction in fact capitalizes on retrieval suppression (and benefits from it). This framework is highly consistent with their findings. So, a directed forgetting instruction (by this view) would engage inhibitory control mechanisms that can suppress both hippocampal and amygdala activity in parallel, creating their effects. Anderson & Floresco. 2022.

3. The speculations in the paper about the neural basis of memory control in the prefrontal cortex

seem largely informed by the item-method directed forgetting literature, which is fine. But by broadening out, you can take advantage of the clear meta-analytic evidence we have developed to isolate memory control in the right DLPFC (mainly MFG) and IFG (e.g. Apsvalka et al. 2022; and Guo et al. 2018; see also, Anderson, Bunce, and Barbas, 2016). This is more likely accurate than the superior frontal gyrus speculation included in the paper and the authors should consider re-evaluating that speculation.

4. The paper presents a somewhat biased recounting of the theoretical account of item-method directed forgetting (in discussion), saying that the effects are “typically” explained by selective rehearsal. In fact, selective rehearsal and inhibitory control are both strongly advocated by different communities, and they are both likely to be true. It would be helpful if they don’t present selective rehearsal as some kind of default.

5. I would highly recommend that the authors check out Anderson & Hulbert (2021) annual review of psychology for our latest accounting of the mechanisms of active forgetting. I think that they will find it interesting and illuminating. I would also advise linking into some of the nice intracranial recording studies that have been done with item method DF, one of which is discussed in that paper. There is a lovely paper by Marie Fellner and Nikolai Axmacher that is also worth a read.

6. I tend to think of item-method directed forgetting as encoding inhibition, a process that relies on inhibitory control of hippocampal activity just after encoding. This process could explain the current effects including the SCR effects, if one simply posits parallel regulation of hippocampus and amygdala, as was observed in Gagnepain et al. 2017 *Jneuro*, but during directed forgetting.

7. In terms of clinical application and its potential, I would like to suggest that the authors consider checking out Mamat & Anderson (2023) *PsyRxiv*, which is a direct attempt at clinical application applying memory control to feared events. This paper is likely to be published this year. Also, Zhu, Anderson & Wang in *Nature Communications* is another interesting extension relevant to clinical application.

Ok, I hope this helps. I think that the authors have a lovely paper here, and I hope that they will make the effort to broaden their focus a bit, as I think it will increase the paper’s impact.

Mike Anderson

RESPONSES TO REVIEWERS

Reviewer 1

1. First, the authors conclude that their results show that forgetting instructions can interfere with emotional associative memory (last sentence of Abstract). However there was no manipulation of the emotional content of the outcome. Of course the use of an innocuous outcome would preclude measurement of skin conductance responses, but it would still be informative to directly test the impact of emotional content on the self-report memory measures. If there was no effect of emotional content on these measures, it would suggest that there is nothing special about emotional memories and that skin conductance might simply be another readout of a memory when the content happens to be emotional.

We agree with the reviewer that there may in fact be nothing special about emotional memories. With emotional associative memories, we are referring to the type of memory created using a fear conditioning procedure in humans. CS+ memories are considered to be emotional, as they are created by repeatedly pairing the CS+ images with an aversive US, yielding a conditioned (fear) response to those cues. CS- memories can be considered safety memories, as they are created by pairing the CS- images with the absence of the aversive US. We did test the effect of emotional content on all our measures (reported as the main effect of stimulus in all our analyses), and for the majority of these analyses, we found significant differences in directed forgetting between CS+ and CS- (SCR, free recall, and liking ratings). That being said, we do not claim that emotional associative memory is special when it comes to directed forgetting. If anything, our main conclusion is that directed forgetting happens even for emotional associative memories, contradicting earlier suggestions that directed forgetting may occur for non-emotional memories only (or to a lesser extent for emotional memories). In that sense, emotional associative memories seem to behave rather similarly as non-emotional memories, and even if measured through a non-verbal measure (see also point 2).

2. Related to the above point, the authors motivate their study by referring to a possible dissociation between “declarative” memory and “emotional” memory, without making it clear what they mean by this distinction outside the narrow field of directed forgetting. In the Discussion, they fail to acknowledge that their different measures were broadly consistent, thus allowing the possibility of a single memory system regardless of content.

Whether there is a single memory system is a topic of debate in the field and beyond the scope of what we would like to present in our manuscript. That being said, we absolutely agree that our results can be consistent with a single memory system perspective. We intended to make a distinction between declarative and non-declarative *read-outs* of memory rather than between declarative and non-declarative memory systems. We agree that the representational structure underlying free recall, recognition, SCR and liking ratings may well be all declarative (propositional) in nature. The crucial contribution that our data make is in demonstrating that even if the content of memory has a clear emotional valence (as when an association is learned between an image and an aversive outcome), a directed

forgetting manipulation can affect the expression of that memory, both in a purely declarative read-out and in a non-declarative, non-verbal read-out.

3. The authors measured both memory for specific CS items as well as for CS-US mappings, and they switch back and forth between the results so that it is sometimes unclear which one they are referring to (e.g. in the discussion of retrieval inhibition on page 18). The question about how these two types of memory relate to each other is an important one. In the context of an associative learning paradigm with multiple presentations of each pairing (3 in this experiment), are we to assume participants have separate episodic memories for each pairing, or do they form a general representation of the pairing on the first trial which is then augmented on subsequent trials? More generally, is there a distinction between learning and memory? It would be good for the authors to address these issues and explain to what extent a memory framework is appropriate to describe what happens in associative learning.

We thank the reviewer for pointing out that the item and associative memory results were not always clearly distinguished. We have revised the manuscript to always include the word “item” or “associative” when discussing these results. Further, on the basis of our results we have to remain agnostic as to whether participants rely on separate episodic memories for each pairing of a given CS with the presence or absence of the US or whether they rely on one representation that is updated or strengthened with each repetition. Both of those possibilities would be compatible with our results; the directed forgetting manipulation could interfere with the formation of specific episodic memories or with the acquisition and strengthening of a non-episodic memory representation. We do think that a memory framework is appropriate to describe what happens in associative learning, whether that memory framework be episodic or not. Surely associative learning will result in changes in knowledge representations in the mind.

4. In the Discussion, I think the authors dismiss the selective rehearsal explanation of their results too readily (top of page 18). The use of multiple trials does not reduce the relevance of rehearsal of CS-US pairs. A rehearsal explanation fits with the procedure of the study more naturally than retrospective inhibition or interference with an established associative memory.

What we intended to convey, is that the repeated presentation of each pairing would mean that physically-induced rehearsal would occur in our procedure, even if mental rehearsal would be reduced under forget instructions. As such, finding directed forgetting effects in our procedure would seem to pose somewhat of a challenge for an explanation in terms of selective rehearsal, also in light of the almost perfect item recognition performance on all trial types (including both CS+F and CS-F). Further, if participants were rehearsing CS-US pairs more, that wouldn't explain the heightened performance on CS-F trials (thus, CS-NoUS pairs) on associative free recall and item/associative recognition performance. That being said, we agree that retrospective inhibition or interference may also represent imperfect explanations for our findings, and we ultimately refrain from drawing definitive conclusions regarding the underlying mechanism of the results observed in our manuscript.

5. Conversely, the logic of attentional inhibition in terms of freeing up cognitive resources (pages 18-19) does not apply readily to the present design, where forget and control pairings were spaced apart by an ITI of 15-10s and hence did not directly compete for processing.

This is a fair point. The reviewer is correct in pointing out that, given the large ITIs, there probably is no competition in working memory. However, attentional inhibition can also be taken to refer to the intentional suppression of goal-irrelevant information (forget trials) and an enhancement of attention to goal-relevant information (remember trials) in the absence of direct competition for resources; neural evidence exists to support such an account (see also points of Reviewer 3). To be fair, we have doubts ourselves as to whether this is indeed the main mechanism responsible for the results we obtained, but on the basis of the data we have collected in these two experiments, we do not feel we have solid arguments to refute it.

6. In Figure 1 shock should be explicitly represented in the diagram for CS+ trials

The shock is represented in Figure 1 at the right upper corner of the CS+ image as a lightning bolt. We increased the size of the lightning bolt to make it more readily noticeable.

7. When was the working memory task given?

The working memory tasks were administered 10 minutes after the conclusion of the DF part of the procedure in Experiment 2 (see p. 28).

8. Page 25: the instructions given to participants presumably referred to “shock” rather than “US”?

The instructions referred to “electrical stimulus”. We do not use the word “shock” in our lab, because it is a loaded term for participants that invites a notion of intensity of stimulation that is far stronger than what we use. We refer to it as the US throughout the manuscript for consistency.

Reviewer 2

Evaluation: The manuscript addresses a highly relevant and timely research question. It is well written and the results and corresponding conclusions to be drawn are clear. I recommend that the manuscript is accepted after a minor revision and detail my comments below.

We appreciate the positive evaluation of our work. Please see below for responses to the individual comments.

1. On page 4, the authors list stimulus-response (S-R) associations as declarative associative memories. However, stimulus-response associations are most often argued to be procedural in nature. Furthermore, recent findings raise some doubt as to whether such procedural stimulus-response associations can be forgotten based on instructions, that is,

whether their behavioral effects, like repetition priming effects, can be affected by forget instructions (e.g., Dames et al., 2022, <https://escholarship.org/uc/item/1s48z1z6>). Of course, it is reasonable to assume that there is also a declarative stimulus-response association that is, for instance, the basis of explicit knowledge regarding the stimulus-response association and can be assessed via recall or recognition performance (rather than behavioral effects). Nevertheless, I suggest that the authors update their corresponding description to better accommodate perspectives on stimulus-response associations that focus on their procedural aspects by, for instance, clarifying that they merely refer to aspects related to the recall/recognition of experienced stimulus-response pairings.

We fully agree with the reviewer regarding the procedural nature of S-R associations and apologize for the oversight – we merely wanted to distinguish between the emotional (fearful) associations created during fear conditioning versus all others. We have now updated the text to read “In recent years, researchers have started investigating DF of declarative and procedural associative memories, employing either unrelated word pairs, scene-object pairs, or arbitrary stimulus-response (S-R) pairings (left/right key presses in response to words), and have observed a DF effect for such associative information as well¹⁴⁻¹⁷, but see¹⁸” (p. 4).

2. As far as I could see, the authors did not explicitly test whether fear conditioning effects still persisted after a forget instruction (i.e., comparison of CS+F and CS-F). I believe that information on these test should be added, as whether forgetting fully abolishes fear conditioning effects or merely reduces them is of theoretical importance and a relevant piece of information when it comes to the discussion of underlying mechanisms. Based on the figures, it appears as if fear conditioning effects persist even after forgetting at least in Experiment 2. Correspondingly, I would like to recommend that the authors also discuss this finding specifically. That is, in the authors’ opinion, can any of the assumed mechanisms be excluded based on the finding that fear conditioning persists even after forgetting?

The reviewer is correct in noting that some fear conditioning persists after a forget instruction in Experiment 2 (but not in Experiment 1), along with the partial preservation of associative memory in declarative measures (free recall and recognition). We have added these additional SCR analyses (see pp. 7-8) to the revised manuscript and discussed them (see p. 16) “It should be noted that some fear conditioning was still maintained under forgetting instructions in Experiment 2, but this is hardly surprising given the robust, immediate learning that transpires during fear conditioning; if anything, it is remarkable that directed forgetting occurred at all during our procedure as the forgetting instructions are working in opposition here to the fact that the repeated pairings of the CSs invite remembering rather than forgetting”. We cannot argue that this would lead to the exclusion of any of the assumed mechanisms described in the previous version of the manuscript, but we did add a novel explanation for our findings in the revised version (see also point 3 below) that may also explain how DF can reduce fear conditioning.

3. The authors’ finding that a higher proportion of associations with CS+ as compared to CS- were forgotten seems to contradict the notion that emotionally-relevant content is

typically remembered better/forgotten less readily. In their discussion, could the authors elaborate a bit more on their thoughts regarding their finding that association recall was worse for CS+ as compared to CS- after forget instructions?

We agree with the reviewer that this observation is surprising. To be fair, any thoughts we could offer are speculative at this point, as we do not have any neural evidence to elucidate what is happening in different brain regions during our procedure and we also did not ask participants for their subjective feelings/ideas/strategies following these first experiments. While not reported in the present manuscript, we did ask participants in later experiments what strategies they used to remember/forget and their responses do not point to any differences in strategies used for CS+ versus CS- trials. However, we have added a possible explanation to the revised manuscript (see pp. 20-21), suggesting that the retrieval stopping model (Anderson & Floresco, 2022) may account for our findings, noting however that this would need to be tested in a neural DF procedure (see also responses to Reviewer 3).

4. Other studies assessing directed forgetting most often use some task (e.g., Corsi task) to purge working memory before testing recall/recognition. Was there a particular reason why this was not done in the present experiments?

Of note, not all DF studies employ such a procedure. Given that these were our first studies in this line of research, we wanted to observe a “clean” DF effect that was not tainted with possible increased cognitive load. We do take the reviewer’s suggestion on board for future experiments, and we have noted the absence of such a task as a limitation in our discussion (see p. 18).

5. I especially appreciate that the authors differentiated between forgetting of the stimulus and forgetting of the association. This aspect could be stated a bit more explicitly in the introduction to further guide the readers and prevent misconceptions that are only resolved later.

We thank the reviewer for this suggestion and agree that our formulation might have not been sufficiently clear. In the revised manuscript, we have pointed to the differences between item and associative information in our introduction, both when describing previous literature (p. 4) “However, in these studies the authors solely measured general recognition performance, and while they observed diminished recognition, in most cases it was impossible to disentangle whether memory deficits were due to an impaired recognition of the items (i.e., disrupted item recognition) or an impaired retention of the association between item pairs (i.e., disrupted associative recognition)” and our own study results (p. 5) “We assessed memory retention for all items and their associations in subsequent free recall and recognition tasks and obtained retrospective liking ratings for all CS stimuli. In line with our hypothesis, we observed impaired item and associative recall, as well as reduced associative recognition, following a forget instruction.”

6. The authors have not shared their analysis scripts and have used several programs for analysis. I would appreciate it if the authors could briefly justify why different analysis programs had to be used and why they did not share their analysis scripts. It would be ideal if they could also share their analysis scripts.

We apologize for the confusion; to be clear, we used only one package for statistical analysis – JASP (which can be downloaded for free on any operating system). This package works with a graphical user interface and does not allow for the generation of script. We did additionally use SPSS, but only to treat the data for missing values using the linear trend at point method; no analyses were performed with SPSS. Upon initial submission, we had already shared our full experiments and full dataset on the OSF project page indicated in the manuscript under “Data availability” (<https://osf.io/65x7a/>). Following the advice of the editor and the current comment of the reviewer, we have now added two additional subsections to our OSF project page: 1) “Analyses” containing all our analysis files from JASP (JASP files are interactive files and include the datasets, full analyses showing which test was performed/which variables were included, and the statistical outputs) and PDF files of the statistical outputs, and 2) “Data underlying manuscript figures” which includes the specific data points that are reflected in each figure of the manuscript.

Reviewer 3

Comments. It was a genuine pleasure to read this paper. I was seriously impressed not only by the quality of the scientific effort, but also by the highly consistent and persuasive findings in SCR and associative measures, greater for CS+ than CS-. It was also made more convincing by the preregistration of the effects. This is certainly the kind of noteworthy paper that should be published in Communications Psychology. And in a highly unusual turn of events, I actually have little to say by way of critical commentary of the work itself, or its interpretation. Statistics are largely appropriate; questions are important and highly relevant to readership of this journal. This was simply a very well-done study—very professional, very careful, and supremely interesting. Well done.

We thank the reviewer for providing such a positive assessment of our manuscript and our work.

On the other hand, I have a bit more to say about the scholarship of the paper. I was struck by the unfortunately narrow focus of the paper on research using the directed forgetting paradigm, with a virtual absence of highly relevant work on memory control derived from the Think/No-Think procedure. The readership would clearly benefit from a synthesis of these bodies of work in this paper and this would need to be done to integrate the finding with the field. Here are some notable points that I would like the authors to address to relate this work on item method directed forgetting to retrieval suppression in the Think/No-Think task:

We absolutely agree that there is a lot of overlap between these two literatures, and we are aware of the innovative work that has been done along these lines for so many years. The reason why we initially chose to keep the literature on memory control in TNT procedures out of the manuscript is that we believe that there might be different processes at play during TNT procedures. In these procedures, participants first complete a learning phase, and then go on to either “think of” or “suppress” those earlier memories, with the instruction appearing at the start of the trial. In DF, participants go through a learning phase and are presented with instructions at trial offset; memory retention is then probed without

concurrent instructions. That being said, we truly appreciate the feedback and have integrated many of the ideas and suggestions below (including the retrieval stopping model as an explanation for our results), which we feel has resulted in a better manuscript.

1. Can Memory Control Processes Disrupt Affective Representations? One might get the sense that this has never been done before from the paper, and this actually is largely true within the confines of the item-method directed forgetting procedure, at least when it comes to (a) psychophysiological indices of emotion and (b) conditioning. However, in work on retrieval suppression (where people are given a cue to a previously associated memory and asked to stop retrieval), it has now been repeatedly found that retrieval suppression affects both subjective and psychophysiological indices of emotion (e.g., Gagnepain et al. 2017, Jneuro; Legrand et al. Scientific reports; Harrington et al. Clinical Psychological Science; Nishiyama & Saito, 2022; Mamat & Anderson 2023, PsyRxiv) and these effects have been tied to regulation of the amygdala. The latter fact seems like it would be helpful in putting a mechanistic face on why the current effects may impact both SCR and associative components relating to fear.

We have now added the retrieval-stopping model and associated explanations to our discussion. It is indeed a possible explanation of our results, given the combined effects on subjective and psychophysiological indices. See pp. 20-21 “In search of an explanation for our findings, one may find inspiration in the work of Anderson and colleagues on retrieval suppression using the Think/No-Think (TNT) paradigm^{30,31}. Extensive research on retrieval suppression has recently led to the proposal of the retrieval stopping model³² of fear extinction, which can be extended to explain our results here. According to this model, when a person encounters a reminder of an unwanted memory, they can terminate (or suppress) the retrieval of that memory by engaging prefrontal cortical regions (most notably the right dorsolateral prefrontal cortex; rDLPFC) that down-regulate the amygdala and suppress further processing in the hippocampus. Notably, retrieval suppression has been shown to affect both subjective and psychophysiological (e.g., SCR, heart rate variability) indices of emotion, and it would allow for a straightforward explanation of our results^{33,34}. Indeed, this model may very well account for the results that we obtained in free recall and recognition, as well as explain why our participants were able to stop the retrieval of the (unwanted) emotional associations. As for the SCR results, one may argue that from the second presentation of each CS onwards, retrieval suppression can prevent the retrieval of the association established through the first CS presentation, thereby impairing SCR expression. As such, this model could provide a plausible explanation of our results. Whether it indeed does, remains however to be tested in future research addressing the neural underpinnings of the current findings.”

2. What might the relationship between fear conditioning and intentional forgetting be? The authors may be interested to learn of our recent attempt to synthesize work on motivated forgetting and fear conditioning. We have a paper in Neuropsychopharmacology with Stan Floresco synthesizing rodent and human literature on this, and have offered the “Retrieval stopping model of fear extinction” in which we posit that fear extinction in fact capitalizes on retrieval suppression (and benefits from it). This framework is highly consistent with their findings. So, a directed forgetting instruction (by this view) would engage inhibitory control mechanisms that can suppress both

hippocampal and amygdala activity in parallel, creating their effects. Anderson & Floresco. 2022.

This is a very interesting framework, and we are very happy to see the two different literatures combined. We have added this as a possible explanation for our results (see also point 1 above).

3. The speculations in the paper about the neural basis of memory control in the prefrontal cortex seem largely informed by the item-method directed forgetting literature, which is fine. But by broadening out, you can take advantage of the clear meta-analytic evidence we have developed to isolate memory control in the right DLPFC (mainly MFG) and IFG (e.g. Apsvalka et al. 2022; and Guo et al. 2018; see also, Anderson, Bunce, and Barbas, 2016). This is more likely accurate than the superior frontal gyrus speculation included in the paper and the authors should consider re-evaluating that speculation.

Indeed, we did use the item-method DF literature to reiterate their neural findings in our discussion. We have now removed these statements from the revised manuscript and refer to the studies mentioned here. We have also mentioned the rDLPFC when introducing the retrieval stopping model (see also point 1 above).

4. The paper presents a somewhat biased recounting of the theoretical account of item-method directed forgetting (in discussion), saying that the effects are “typically” explained by selective rehearsal. In fact, selective rehearsal and inhibitory control are both strongly advocated by different communities, and they are both likely to be true. It would be helpful if they don’t present selective rehearsal as some kind of default.

We certainly agree with this comment, and we regret that our statements seemed biased. We have now rephrased the introduction of these mechanisms in the discussion to state (pp. 19-20): “item-method DF effects have often been attributed to selective rehearsal”; “Another candidate for explaining DF effects, that has been equally advocated in the literature, is retrieval inhibition”; “A third popular account, attentional inhibition...”.

5. I would highly recommend that the authors check out Anderson & Hulbert (2021) annual review of psychology for our latest accounting of the mechanisms of active forgetting. I think that they will find it interesting and illuminating. I would also advise linking into some of the nice intracranial recording studies that have been done with item method DF, one of which is discussed in that paper. There is a lovely paper by Marie Fellner and Nikolai Axmacher that is also worth a read.

We appreciate the pointer to Anderson & Hulbert (2021), which brings together many different varieties of active forgetting and inspires many ideas to be tested in the future. However, we have not added the intracranial recording studies to our revised manuscript as we believe that they are (conceptually) too far removed from the (solely) behavioral data that we have in our manuscript.

6. I tend to think of item-method directed forgetting as encoding inhibition, a process that relies on inhibitory control of hippocampal activity just after encoding. This process could explain the current effects including the SCR effects, if one simply posits parallel regulation

of hippocampus and amygdala, as was observed in Gagnepain et al. 2017 Jneuro, but during directed forgetting.

We also very much favor this idea of item-method DF, even though we also speculate (last paragraph of the discussion) that there could be inhibition during retrieval as well. This clearly remains to be tested in a neural DF fear-conditioning procedure.

7. In terms of clinical application and its potential, I would like to suggest that the authors consider checking out Mamat & Anderson (2023) PsyRxiv, which is a direct attempt at clinical application applying memory control to feared events. This paper is likely to be published this year. Also, Zhu, Anderson & Wang in Nature Communications is another interesting extension relevant to clinical application.

We appreciate the pointers to further literature and have made reference to both of these papers in our discussion (see p. 21) “Along those lines, studies using memory suppression techniques have already provided some primary evidence demonstrating reduced recall of unwanted memories following subliminal memory reactivation⁴⁷ and reduced (self-reported) symptoms of anxiety, negative affect, and depression, following a 3-day, online suppression training⁴⁸”.

1st Aug 23

Dear Dr Chalkia,

Your manuscript titled "Disrupting emotional associative memory through directed forgetting" has now been seen by our reviewers, whose comments appear below. In light of their advice I am delighted to say that we are happy, in principle, to publish a suitably revised version in Communications Psychology under the open access CC BY license (Creative Commons Attribution v4.0 International License).

Reviewers #2 and #3 from the previous round are satisfied that all their concerns were addressed. Reviewer #1 lists a few remaining concerns. In detail, they ask for some further clarification and discussion; we require you to address all of their comments in a final comprehensive revision. Please provide a Response to the Reviewers document in which you detail the changes you make in revision.

When revising your manuscript, please ensure you remove any speculations and claims of novelty. Further, please highlight any shortcomings in a separate 'Limitations' section (see the attached editorial request table for details).

We therefore invite you to revise your paper one last time to address the remaining concerns of our reviewers and a list of editorial requests. At the same time we ask that you edit your manuscript to comply with our format requirements and to maximise the accessibility and therefore the impact of your work.

EDITORIAL REQUESTS:

SUBMISSION INFORMATION:

OPEN ACCESS:

Communications Psychology is a fully open access journal. Articles are made freely accessible on publication under a [CC BY license](http://creativecommons.org/licenses/by/4.0) (Creative Commons Attribution 4.0 International License). This license allows maximum dissemination and re-use of open access materials and is preferred by many research funding

bodies.

For further information about article processing charges, open access funding, and advice and support from Nature Research, please visit <https://www.nature.com/commspsychol/article-processing-charges>

At acceptance, you will be provided with instructions for completing this CC BY license on behalf of all authors. This grants us the necessary permissions to publish your paper. Additionally, you will be asked to declare that all required third party permissions have been obtained, and to provide billing information in order to pay the article-processing charge (APC).

* **DATA AVAILABILITY:**

[link redacted]

Best regards,

Antonia Eisenkoeck

Antonia Eisenkoeck
Senior Editor
Communications Psychology

REVIEWERS' COMMENTS:

Reviewer #1 (Remarks to the Author):

I'm pleased the authors have clarified whether they are talking about item memory vs associative memory throughout the revised ms. However, they haven't made any changes to the ms in relation to any of the theoretical/interpretive issues I raised. This is a pity, because I think that generalist readers may be interested in their views on these issues. More importantly, by avoiding discussion of important theoretical implications of their findings, they have missed an opportunity to increase the impact of their paper. I would encourage the authors to tackle these issues in the paper itself. I have made some comments on their response document below in case these are helpful.

1. Whether the results are specific to emotional memory

The comparison between CS+ and CS- does not speak uniquely to the question of emotional content because it is confounded by whether the stimulus predicts the presence or absence of the outcome (US). That is, there could well be a difference between the impact of directed forgetting on CS+ and CS- even in a task where the outcome is neutral. Conversely, CS- in a fear conditioning paradigm does have emotional content – it conveys safety or relief. Because the authors have not manipulated the emotional relevance of the outcome, they can't say one way or the other whether their results are specific to associative learning with emotionally significant outcomes. However, if they think that their result is important because they observed directed forgetting despite using an emotionally significant outcome, they should make this clear in the paper.

2. Use of the terms “declarative” and “emotional/procedural”

The authors state that the topic of whether there is a single memory system is beyond the scope of their ms. So why describe their study in terms of the declarative-emotional distinction in the first place? In the revision they have left unchanged the text on page 4 that indicates they are using the terms declarative and emotional/procedural in the conventional way to refer to the nature of the memories themselves, not the read-out as they argue in their response document.

3. Item vs associative memory

Similarly, the authors might prefer to remain agnostic about the relationship between the memories for individual trials and associative knowledge, but it is hard to avoid the issue when the one of the distinctive features of their study is that it involves CS-US pairings and each pairing is presented on multiple trials. At the very least this should be raised in the Discussion.

4. Plausibility of selective rehearsal account of the findings

Physically presenting pairs of stimuli by no means ensures the relationship between them will be encoded or rehearsed, as this is quite distinct from encoding of the individual stimuli. The impact of the directed forgetting manipulation on CS- associative performance could potentially be explained by noting that the absence of a significant stimulus such as shock is an important outcome in its own right. To show high accuracy in categorizing the CS-s, participants would need to devote as much effort to rehearsing the CS- -> no shock pairings as to the CS+ -> shock pairings.

5. Lack of competition for cognitive resources

In their response letter, the authors agree there is little competition in working memory and go on to propose a different mechanism for attentional inhibition (which sounds a lot like reduced rehearsal). However, they have left unchanged their description of attentional inhibition in terms of maintaining cognitive resources to process R items in the revised ms (p. 20).

Reviewer #2 (Remarks to the Author):

I thank the authors for their detailed replies to my comments. They have effectively addressed my concerns and I applaud them for this fine piece of work.

Reviewer #3 (Remarks to the Author):

The authors have responded to the comments that I made concerning the original submission and I am happy with the changes they made.

One minor point---at one point, when discussing the impact of suppression on heart rate measures, they use the term "heart rate variability" when they should instead say "heart rate deceleration" which are two, very different measures. Please fix that.

Congratulations on a nice article.

Mike Anderson

RESPONSES TO REVIEWERS

Reviewer 1

I'm pleased the authors have clarified whether they are talking about item memory vs associative memory throughout the revised ms. However, they haven't made any changes to the ms in relation to any of the theoretical/interpretive issues I raised. This is a pity, because I think that generalist readers may be interested in their views on these issues. More importantly, by avoiding discussion of important theoretical implications of their findings, they have missed an opportunity to increase the impact of their paper. I would encourage the authors to tackle these issues in the paper itself. I have made some comments on their response document below in case these are helpful.

We thank the reviewer for providing additional guidance to help us improve our manuscript. Please see below for detailed changes and replies to each comment.

1. Whether the results are specific to emotional memory. The comparison between CS+ and CS- does not speak uniquely to the question of emotional content because it is confounded by whether the stimulus predicts the presence or absence of the outcome (US). That is, there could well be a difference between the impact of directed forgetting on CS+ and CS- even in a task where the outcome is neutral. Conversely, CS- in a fear conditioning paradigm does have emotional content – it conveys safety or relief. Because the authors have not manipulated the emotional relevance of the outcome, they can't say one way or the other whether their results are specific to associative learning with emotionally significant outcomes. However, if they think that their result is important because they observed directed forgetting despite using an emotionally significant outcome, they should make this clear in the paper.

Prior to conducting the experiments included in this manuscript, we completed a few more experiments that did not include any psychophysiology. In one of those experiments, we presented both CS+ and CS- items with an outcome: CS+ items were paired with a negatively valenced image and CS- items were paired with a neutral image, thus, all stimuli predicted an outcome, either emotional or neutral. We observed the same results, with larger effects on the CS+ items. Given that these data are not included here, we cannot make this argument in the manuscript, but indeed, we do believe our results are important because we observed DF despite using an emotionally significant outcome, and in a non-verbal readout, which is counterintuitive to what one would expect when considering prior literature. We have clarified this point in the introduction (see pp. 4-5): “Yet these DF effects on declarative memory performance, as probed through verbal report, may have little bearing on the expression of emotional associative memory, considering that the expression of emotional memory is often more resistant to interference and that it can be expressed through automatic reactions (e.g., psychophysiological responses). Indeed, DF procedures utilizing (single-item) emotional words or images have yielded weak DF effects in verbal report at best, with item recall and/or recognition for emotional stimuli consistently higher than for neutral stimuli. No research has addressed the ability of DF to reduce non-verbal memory expression (e.g., psychophysiological responding), despite the central role of non-voluntary retrieval of associative memory information in emotional disorders” and in the discussion (see p. 27): “Our findings are in line with decades worth of prior DF research using neutral stimuli, but they are in contrast with the results of studies that tested non-associative negative emotional stimuli. Such studies reported higher recall/recognition of negative stimuli, which,

consequently, yielded diminished DF effects (but see for an investigation of different categories of emotional stimuli). Further, a recent meta-analysis of item-method DF confirmed that emotional memories are often more resilient to forgetting than neutral memories, but several moderating factors were also reported. In our data, negative valence did not weaken the DF effect, as is reflected in the robust disruption of CS+ retention in both verbal and non-verbal indices of fear memory. Thus, with the current design, we demonstrate that it is indeed possible to interfere with memory for emotional information through DF and bring additional evidence to the field.”

2. Use of the terms “declarative” and “emotional/procedural”. The authors state that the topic of whether there is a single memory system is beyond the scope of their ms. So why describe their study in terms of the declarative-emotional distinction in the first place? In the revision they have left unchanged the text on page 4 that indicates they are using the terms declarative and emotional/procedural in the conventional way to refer to the nature of the memories themselves, not the read-out as they argue in their response document.

In our introduction, we are merely reviewing the literature, and the papers we refer to are discussing their findings using the terminology “declarative” and “procedural” because this is a distinction that is very commonly found in the literature. We do not mean to argue for or against a single memory system because we do not have data to support either side from the two experiments included in this manuscript. That being said, in the revised manuscript, we have now made sure to point to “read-outs” of memories, or to their “expression”, or to “memory performance” (see p. 4): “In recent years, researchers have started investigating DF of declarative and procedural read-outs of associative memories, employing either unrelated word pairs, scene-object pairs, or arbitrary stimulus-response (S-R) pairings (left/right key presses in response to words)...” “Yet these DF effects on declarative memory performance, as probed through verbal report, may have little bearing on the expression of emotional associative memory, considering that the expression of emotional memory is often more resistant to interference and that it can be expressed through automatic reactions (e.g., psychophysiological responses).” We also indicate in multiple places that emotional memories, as we refer to them throughout the manuscript, can be expressed through non-verbal, automatic, psychophysiological responses, which we consider to be an emotional read-out of memory. Last, we repeatedly distinguish between declarative (free recall and recognition) and physiological (SCR) indices of memory expression when discussing our findings.

3. Item vs associative memory. Similarly, the authors might prefer to remain agnostic about the relationship between the memories for individual trials and associative knowledge, but it is hard to avoid the issue when the one of the distinctive features of their study is that it involves CS-US pairings and each pairing is presented on multiple trials. At the very least this should be raised in the Discussion.

In the revised manuscript, we have added a discussion point relating to this matter. Please see p. 25: “Regarding the nature of the associative memories being created with the current design, we have to remain agnostic as to whether participants rely on separate episodic memories for each pairing of a given CS with the presence or absence of the US or whether they rely on one representation that is updated or strengthened with each repetition. Both of those possibilities would be compatible with our results; the directed forgetting manipulation could interfere with the formation of specific episodic memories or with the acquisition and

strengthening of a non-episodic memory representation.”

4. Plausibility of selective rehearsal account of the findings. Physically presenting pairs of stimuli by no means ensures the relationship between them will be encoded or rehearsed, as this is quite distinct from encoding of the individual stimuli. The impact of the directed forgetting manipulation on CS- associative performance could potentially be explained by noting that the absence of a significant stimulus such as shock is an important outcome in its own right. To show high accuracy in categorizing the CS-s, participants would need to devote as much effort to rehearsing the CS- -> no shock pairings as to the CS+ -> shock pairings.

We certainly agree with the reviewer and never claim in our manuscript that rehearsal was not taking place at all. On the contrary, given the almost perfect performance in item recognition for all stimuli, rehearsal must have been taking place, for all stimuli, but it was not selective to the remember items as theory would suggest, and that is why we believe that selective rehearsal was not the mechanism solely responsible for our results. We have rephrased this part of our discussion in the revised manuscript to clarify that rehearsal was taking place, see p. 28: “According to this account, items that are presented one by one are held in working memory until the instruction to remember or forget is introduced; participants are then thought to continue actively rehearsing the R items, while stopping to rehearse F items, which eventually passively decay. It might be difficult to maintain this explanation for the findings of our experiments, where all items were repeated, and thus, additional rehearsal of both R and F items should have occurred (due to mere repetition), be it perhaps to a lesser extent for F items than for R items (due to the F cues halting rehearsal). However, we observed ceiling performance in item recognition across experiments, which suggests that rehearsal was indeed actively taking place, but it was not selective to remember items only. Nonetheless, future studies could introduce variable delays between CS presentations and instruction cues or a concurrent taxing working memory task to see if disrupting rehearsal can diminish the DF effect, in support of the selective rehearsal hypothesis.”

5. Lack of competition for cognitive resources. In their response letter, the authors agree there is little competition in working memory and go on to propose a different mechanism for attentional inhibition (which sounds a lot like reduced rehearsal). However, they have left unchanged their description of attentional inhibition in terms of maintaining cognitive resources to process R items in the revised ms (p. 20).

We apologize for this oversight and have updated the description of attentional inhibition in the revised manuscript, see p. 29: “A third popular account, attentional inhibition, proposes that upon presentation of an F instruction, attentional mechanisms are engaged that actively suppress the processing of information that has become goal-irrelevant (i.e., F items). Attentional inhibition has also been suggested to be responsible for suppressing any goal-irrelevant information that may have entered working memory and for preventing individuals’ attention from returning to information that was previously deemed goal-irrelevant (e.g., CS stimuli following an F instruction in our case).”

Reviewer 2

I thank the authors for their detailed replies to my comments. They have effectively addressed my concerns and I applaud them for this fine piece of work.

We thank the reviewer for their kind words and for providing constructive comments that helped improve our manuscript.

Reviewer 3

The authors have responded to the comments that I made concerning the original submission and I am happy with the changes they made. One minor point---at one point, when discussing the impact of suppression on heart rate measures, they use the term "heart rate variability" when they should instead say "heart rate deceleration" which are two, very different measures. Please fix that. Congratulations on a nice article.

We have made this final change and we are very grateful for all the helpful advice that has allowed us to improve our manuscript.